# Unveiling the key factor for the phase reconstruction and exsolved metallic particle distribution in perovskites

Hyunmin Kim[1,6], Chaesung Lim[2,6], Ohhun Kwon[3,6], Jinkyung Oh[1], Matthew T. Curnan[2], Hu Young Jeong[ID][4], Sihyuk Choi[5✉], Jeong Woo Han[ID][2✉] & Guntae Kim[ID][1✉]

To significantly increase the amount of exsolved particles, the complete phase reconstruction from simple perovskite to Ruddlesden-Popper (R-P) perovskite is greatly desirable. However, a comprehensive understanding of key parameters affecting the phase reconstruction to R-P perovskite is still unexplored. Herein, we propose the Gibbs free energy for oxygen vacancy formation in $Pr_{0.5}(Ba/Sr)_{0.5}TO_{3-\delta}$ (T = Mn, Fe, Co, and Ni) as the important factor in determining the type of phase reconstruction. Furthermore, using in-situ temperature & environment-controlled X-ray diffraction measurements, we report the phase diagram and optimum '$x$' range required for the complete phase reconstruction to R-P perovskite in $Pr_{0.5}Ba_{0.5-x}Sr_xFeO_{3-\delta}$ system. Among the $Pr_{0.5}Ba_{0.5-x}Sr_xFeO_{3-\delta}$, $(Pr_{0.5}Ba_{0.2}Sr_{0.3})_2FeO_{4+\delta}$ – Fe metal demonstrates the smallest size of exsolved Fe metal particles when the phase reconstruction occurs under reducing condition. The exsolved nano-Fe metal particles exhibit high particle density and are well-distributed on the perovskite surface, showing great catalytic activity in fuel cell and syngas production.

[1] School of Energy and Chemical Engineering, Ulsan National Institute of Science and Technology (UNIST), Ulsan 44919, Republic of Korea. [2] Department of Chemical Engineering, Pohang University of Science and Technology (POSTECH), Pohang 37673, Republic of Korea. [3] Department of Chemical and Biomolecular Engineering, University of Pennsylvania, Philadelphia, PA 19104, USA. [4] Department of Materials Science and Engineering and UNIST Central Research Facilities (UCRF), Ulsan National Institute of Science and Technology (UNIST), Ulsan 44919, Republic of Korea. [5] Department of Mechanical Engineering (Aeronautics, Mechanical and Electronic Convergence Engineering), Kumoh National Institute of Technology, Gyeongbuk 39177, Republic of Korea. [6] These authors contributed equally: Hyunmin Kim, Chaesung Lim and Ohhun Kwon. ✉email: sh.choi@kumoh.ac.kr; jwhan@postech.ac.kr; gtkim@unist.ac.kr

Tailoring the functionality of perovskite oxides ($ABO_3$) by decorating the surface with catalytically active particles plays an important role in energy-related applications such as fuel cells, electrolysis cells, metal-air batteries, and supercapacitors[1–6]. The catalyst particles are typically prepared by deposition techniques (e.g. infiltration, chemical vapor deposition, and pulsed laser deposition), in which the catalysts are embedded onto the surface from external precursors[7–9]. However, these techniques require redundant heat-treatments for preparation and the catalyst particles suffer from agglomeration and/or coarsening over time, resulting in performance degradation[10,11]. In this respect, it is of great importance to develop more robust and time-efficient catalyst preparation method. Exsolution phenomenon on the basis of in-situ growth of metal particles has been suggested as an advanced approach to designing perovskite matrix with electro-catalytically active particles[12,13]. In this approach, catalytically-active metal elements (e.g. Pd, Ru, Co, Ni, and Fe, etc…) are initially incorporated into the B-site of perovskite oxides, and then exsolved as metallic particles from the perovskite support under reducing atmosphere[14,15]. As compared with the conventional catalyst preparation methods, the in-situ exsolution process provides benefits of time-efficient catalyst preparation, enhanced catalyst lifetime, and robust thermal stability[16,17]. Notwithstanding the advantages, two major thresholds hinder the practical application of the exsolution process: (i) restricted diffusion of catalytically active cations to the surface due to preferential segregation within the bulk[18], and (ii) structural destruction and/or insulating phase evolution after excessive cation defect formation[19].

In order to address the challenges of the exsolution phenomenon, A-site deficient perovskites (A/B < 1) has been extensively employed as an attractive methodology[2,20–22]. In A-site deficient perovskites, formation of oxygen vacancies is promoted by phase stabilization from non-stoichiometric perovskite to defect-free perovskite under reducing condition, facilitating the B-site exsolution[23,24]. Hence, the B-site exsolution level is proportional to A-site deficiency range ('$\alpha$' for $A_{1-\alpha}BO_{3-\delta}$). Meanwhile, there exists restriction in the variation of A-site deficiency range (about $0 < \alpha < 0.2$ for $A_{1-\alpha}BO_{3-\delta}$) because excessive A-site deficiency may be accompanied by formation of undesirable A-site oxide phases[19,25]. Given these aspects, an alternative corresponding method to further trigger the B-site exsolution is using the in-situ phase reconstruction from simple perovskite to Ruddlesden-Popper (R-P) perovskite oxides ($A_{n+1}B_nO_{3n+1}$ with n = 1, 2, and 3) via reduction process[26,27]. This strategy facilitates abundant formation of oxygen vacancies during the phase reconstruction, breaking the bottleneck of exsolution capability.

$$ABO_3 \xrightarrow[\text{After reduction}]{} \frac{1}{2}A_2BO_4 + \frac{1}{2}B + \frac{1}{2}O_2 \qquad (1)$$

From Eq. 1, it is presumable that considerable number of cations at B-site will be reduced into metals without A-site segregation after phase reconstruction from simple perovskite ($ABO_3$) to n = 1 R-P perovskite ($A_2BO_4$). Although several perovskites have exhibited superior distribution of catalyst particles on the surface via phase transition to R-P perovskite[28–30], the comprehensive understanding of key factors modulating the phase reconstruction to R-P perovskite is still an open question.

Inspired by the above perspectives, the goal of this study is to identify the significant factor contributing to the phase reconstruction from simple perovskite to R-P perovskite. Here, we systematically report the Gibbs free energy for oxygen vacancy formation ($G_{vf-O}$) of perovskite oxides with various cations as the unprecedented factor affecting the phase reconstruction. The type of phase reconstruction could be predicted with the $G_{vf-O}$ value from PrO and $TO_2$ networks in $Pr_{0.5}(Ba/Sr)_{0.5}TO_{3-\delta}$ (T = Mn, Fe, Co, and Ni), in which the most appropriate cations for the complete reconstruction to R-P perovskite are determined. Afterwards, the phase diagram from in-situ temperature and environment-controlled X-ray diffraction (XRD) measurements reveals the phase reconstruction tendency of $Pr_{0.5}Ba_{0.5-x}Sr_x FeO_{3-\delta}$ (x = 0, 0.1, 0.2, 0.3, 0.4, and 0.5, abbreviated as PBSF in Supplementary Table 1) materials with respect to 'x' value and reduction temperature. Furthermore, the as-exsolved Fe metal particle size and distribution for PBSF after reduction process are observed from microscopy analysis. In accordance with the theoretical calculations and experimental data, $Pr_{0.5}Ba_{0.2}Sr_{0.3}FeO_{3-\delta}$ (A-PBSF30) is adopted as the optimized electrode material for symmetrical solid oxide cell (S-SOC) and demonstrates exceptional electrochemical performance (1.23 W cm$^{-2}$ at 800 °C under fuel cell mode and −1.62 A cm$^{-2}$ at 800 °C under co-electrolysis mode).

## Results

**Density functional theory calculations**. The complete phase reconstruction from simple perovskite ($ABO_3$) to R-P perovskite ($A_2BO_4$) via reduction is considered as one of the efficient strategies to significantly boost the population of exsolved particles. However, the key factors contributing to the phase reconstruction to R-P perovskite has not been investigated. To determine the unexplored factor for the phase reconstruction for the first time, the Gibbs free energy for oxygen vacancy formation ($G_{vf-O}$) and the oxygen vacancy formation energies ($E_{vf-O}$) from the surface AO (A-site) and $BO_2$ (B-site) networks were calculated for $Pr_{0.5}Ba_{0.5}TO_{3-\delta}$ and $Pr_{0.5}Sr_{0.5}TO_{3-\delta}$ (T = Mn, Fe, Co, and Ni) perovskite oxides (Fig. 1 and Supplementary Fig. 1)[31–34]. For the perovskite oxides to undergo phase reconstruction without phase decomposition under reducing condition, the A-site $G_{vf-O}$ value should be positive (A-site $G_{vf-O} > 0$ eV). Moreover, the B-site $G_{vf-O}$ value would be an important factor for determining the type of phase reconstruction. For instance, the B-site $G_{vf-O}$ should be in the range of about −1.2 to 0 eV (−1.2 eV < B-site $G_{vf-O} < 0$ eV) to demonstrate complete phase reconstruction to R-P perovskite in the reduction environment. Considering the aforementioned results and the experimental data, only $Pr_{0.5}Sr_{0.5}MnO_{3-\delta}$ (PSM) and $Pr_{0.5}Sr_{0.5}FeO_{3-\delta}$ (A-PBSF50) are the possible candidates for the complete phase reconstruction to R-P perovskite in this study (Supplementary Fig. 2). Among the two potential candidates, we adopted Fe cation as the more suitable B-site cation because of its much superior catalytic activity for fuel oxidation reaction rather than Mn cation[18]. Accordingly, we systematically analyzed the phase reconstruction tendency of $Pr_{0.5}Ba_{0.5-x}Sr_xFeO_{3-\delta}$ (x = 0, 0.1, 0.2, 0.3, 0.4, and 0.5, abbreviated as PBSF in Supplementary Table 1) materials with respect to different $Ba^{2+}/Sr^{2+}$ ratio.

**Structural characterization**. Before examining the phase reconstruction tendency of PBSF, the crystalline structures after heat-treated in two different environmental conditions were analyzed by X-ray diffraction (XRD) and Rietveld refinement profiles (Supplementary Figs. 3, 4 and Supplementary Table 2). The air-sintered PBSF are all corresponded to simple perovskite structure without detectable secondary phases. On the other hand, after reduction in $H_2$ atmosphere, the PBSF samples were surprisingly changed to different types of phases depending on the $Sr^{2+}$ concentration. As shown in Supplementary Fig. 4b, $Pr_{0.5}Ba_{0.5}FeO_{3-\delta}$ (A-PBSF00), $Pr_{0.5}Ba_{0.2}Sr_{0.3}FeO_{3-\delta}$ (A-PBSF30), and $Pr_{0.5}Sr_{0.5}FeO_{3-\delta}$ (A-PBSF50) were changed to $Pr_{0.5}Ba_{0.5}FeO_{3-\delta}$ – Fe metal & Pr oxide (R-PBSF00), $(Pr_{0.5}Ba_{0.2}Sr_{0.3})_2FeO_{4+\delta}$ – Fe metal (R-PBSF30), and $(Pr_{0.5}Sr_{0.5})_2FeO_{4+\delta}$ – Fe metal (R-PBSF50), respectively. Only catalytically active Fe metal peaks along with complete phase reconstruction to R-P perovskite are

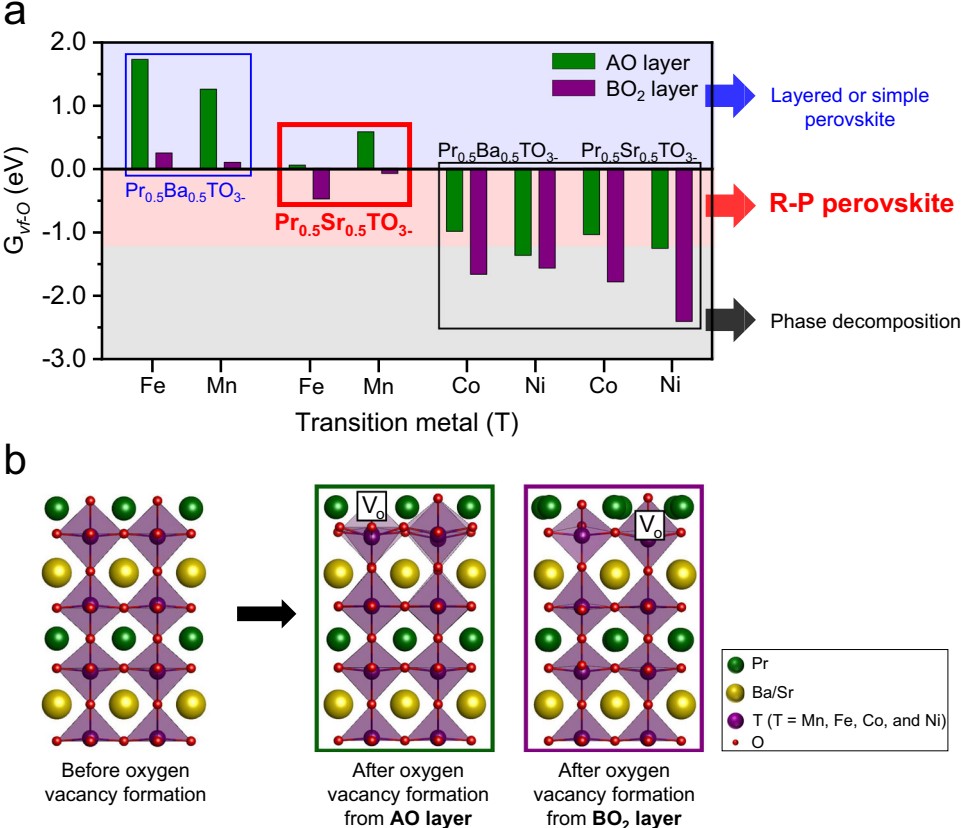

**Fig. 1 Density functional theory calculations. a** Calculated Gibbs free energy for oxygen vacancy formation ($G_{vf-O}$) of $Pr_{0.5}(Ba/Sr)_{0.5}TO_{3-\delta}$ (T = Mn, Fe, Co, and Ni) from the surface AO (green bar) and $BO_2$ (purple bar) networks and the predicted phase change under reducing condition. **b** Schematic illustration of the most stable structure configurations of $Pr_{0.5}(Ba/Sr)_{0.5}TO_{3-\delta}$ (T = Mn, Fe, Co, and Ni) slab models used for the calculations of oxygen vacancy formation energy values from AO and $BO_2$ networks.

observed for R-PBSF30 and R-PBSF50, while R-PBSF00 shows Fe metal and Pr oxide segregation without phase reconstruction under reducing condition. Based on the further Rietveld refinement analysis in Supplementary Fig. 5, R-PBSF30 clearly exhibits the complete phase reconstruction to R-P perovskite with tetragonal structure (space group I4/mmm with lattice parameters of a = b = 3.879 and c = 12.704 Å). The complete phase reconstruction could be also described by Eq. (2), of which considerable amounts of Fe metal are expected to be exsolved in the reduction environment.

$$Pr_{0.5}Ba_{0.2}Sr_{0.3}FeO_3 \xrightarrow[\text{After reduction}]{} \frac{1}{2}(Pr_{0.5}Ba_{0.2}Sr_{0.3})_2FeO_4 + \frac{1}{2}Fe + \frac{1}{2}O_2$$

(2)

**Phase reconstruction tendency analysis from phase diagram.** To precisely analyze the phase reconstruction tendency for $Pr_{0.5}Ba_{0.5-x}Sr_xFeO_{3-\delta}$ (x = 0, 0.1, 0.2, 0.25, 0.3, 0.4, and 0.5) materials, in-situ XRD measurements were systematically conducted in various reduction temperatures and $Sr^{2+}$ concentrations. (Fig. 2a and Supplementary Fig. 6). Figure 2b displays the proposed phase diagram and the corresponding plotted points after in-situ XRD measurements in $H_2$ with elevating temperature intervals of 10 °C. The A-PBSF00 sample remained simple perovskite structure for all reduction temperature range and co-segregation of Fe metal and Pr oxide was observed simultaneously at the reduction temperature higher than 840 °C (Region II in Fig. 2b). Even though there was noticeable phase reconstruction for all $Sr^{2+}$-doped samples,

complete phase reconstruction to R-P perovskite was not accomplished for $Pr_{0.5}Ba_{0.4}Sr_{0.1}FeO_{3-\delta}$ (A-PBSF10), $Pr_{0.5}Ba_{0.3}Sr_{0.2}FeO_{3-\delta}$ (A-PBSF20), and $Pr_{0.5}Ba_{0.25}Sr_{0.25}FeO_{3-\delta}$ (A-PBSF25) even at the reduction temperature of 870 °C. (Region III in Fig. 2b). On the contrary, complete phase reconstruction to R-P perovskite is observed for A-PBSF30, $Pr_{0.5}Ba_{0.1}Sr_{0.4}FeO_{3-\delta}$ (A-PBSF40), and A-PBSF50 at the reduction temperature of approximately 850 °C (Region IV in Fig. 2b). These results indicate that the 'x' value in PBSF should be at least approximately 0.3 along with the reduction temperature of about 850 °C to accomplish complete phase reconstruction, as illustrated in Fig. 2c.

**Effect of $Sr^{2+}$ concentration on phase reconstruction and exsolution.** The role of $Sr^{2+}$ concentration in PBSF in terms of phase reconstruction tendency to R-P perovskite was additionally explored using density functional theory (DFT) calculations. Figure 2d shows the required total energies for the phase reconstruction ($E_{recon}$) from simple perovskite to R-P perovskite of four model structures with different $Ba^{2+}/Sr^{2+}$ ratio. The $E_{recon}$ decreases with increasing $Sr^{2+}$ concentration in PBSF, indicating that the incorporation of $Sr^{2+}$ into $Ba^{2+}$ site promotes the phase reconstruction to R-P perovskite (Supplementary Fig. 7). Furthermore, the $E_{vf-O}$ of four simple perovskite models were calculated in Fig. 2e. More negative $E_{vf-O}$ value implies easier formation of oxygen vacancies in the reduction environment[35]. The $E_{vf-O}$ value becomes more negative after doping $Sr^{2+}$ into $Ba^{2+}$, revealing that the $Sr^{2+}$ doping facilitates the formation of oxygen vacancies in the reduction atmosphere. This trend could be

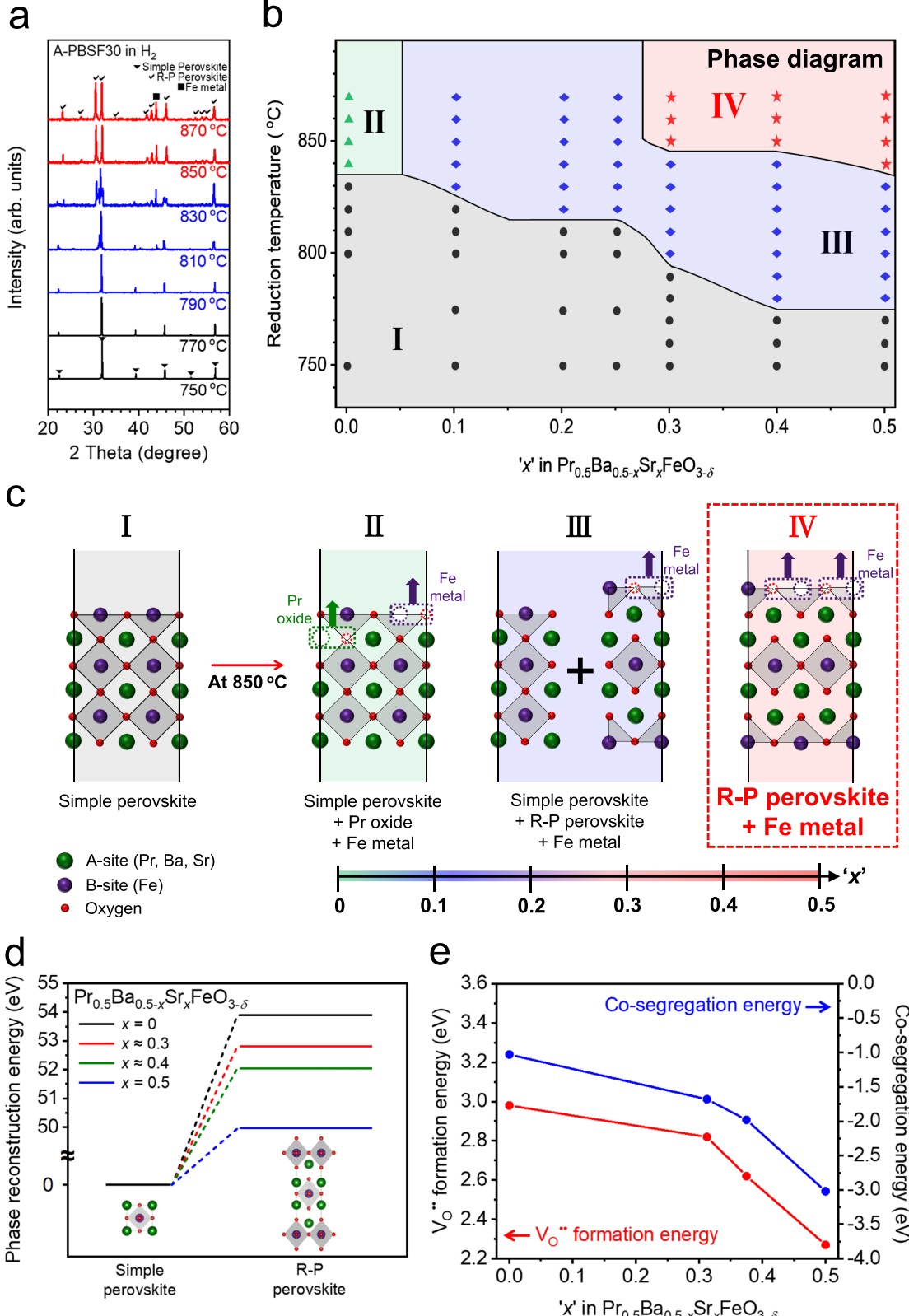

elucidated by the decrease in tolerance factor after replacement of $Ba^{2+}$ by $Sr^{2+}$ (Supplementary Table 3)[36]. A co-segregation energy ($E_{co\text{-}seg}$) associated to the degree of exsolution for B-site transition metal cations under reducing condition was also calculated (Fig. 2e). Interestingly, $E_{co\text{-}seg}$ decreased as the $Sr^{2+}$ contents increased, suggesting the enhanced degree of Fe exsolution with increasing $Sr^{2+}$ concentration.

**Transmission electron microscopy analysis.** On the basis of proposed phase diagram and DFT calculations, the A-PBSF30, the minimum $Sr^{2+}$-doped sample demonstrating complete reconstruction to R-P perovskite, is selected as the target material for structural analysis. The transmission electron microscopy (TEM) and scanning TEM (STEM) analysis were successfully performed to visually probe the complete phase reconstruction

**Fig. 2 Examination of phase reconstruction behavior for $Pr_{0.5}Ba_{0.5-x}Sr_xFeO_{3-\delta}$ material under reducing condition. a-c** Phase reconstruction tendency of $Pr_{0.5}Ba_{0.5-x}Sr_xFeO_{3-\delta}$ material ($x$ = 0, 0.1, 0.2, 0.25, 0.3, 0.4, and 0.5). **a** In-situ powder X-ray diffraction (XRD) patterns of $Pr_{0.5}Ba_{0.2}Sr_{0.3}FeO_{3-\delta}$ (A-PBSF30) under $H_2$ environment. **b** Proposed phase diagram of $Pr_{0.5}Ba_{0.5-x}Sr_xFeO_{3-\delta}$ material ($x$ = 0, 0.1, 0.2, 0.25, 0.3, 0.4, and 0.5) in $H_2$ environment as functions of reduction temperature and $Sr^{2+}$ concentration from in-situ XRD measurements. The phases for region I (gray), II (green), III (blue), and IV (red) are simple perovskite, simple perovskite + Pr oxide + Fe metal, simple perovskite + Ruddlesden-Popper (R-P) perovskite + Fe metal, and R-P perovskite + Fe metal, respectively. **c** Schematic illustration of the above phase diagram. **d, e** Density functional theory (DFT) calculations. Calculated profiles of **d** the relative total energy required for the phase reconstruction from simple perovskite to R-P perovskite and **e** oxygen vacancy formation energies and co-segregation energies as a function of $Sr^{2+}$ concentration in four models.

from simple perovskite to R-P perovskite of A-PBSF30 material (Fig. 3 and Supplementary Fig. 8). From the high-resolution TEM images and corresponding fast-Fourier transformed (FFT) patterns, the lattice spaces between planes of A-PBSF30 and R-PBSF30 are 0.395 nm (Fig. 3a) and 0.634 nm (Fig. 3d), which are consistent with the lattice constant of (001) plane for simple perovskite and the lattice constant of (002) plane for R-P perovskite, respectively. Furthermore, the atomic-scale observations of A-PBSF30 and R-PBSF30 were definitely validated from high-angle annular dark-field (HAADF) STEM images, of which only technically elusive [100] direction is mandatory for R-P perovskite. The locations of cations are well-matched with the simple perovskite (Fig. 3b) and R-P perovskite (Fig. 3e) because the atomic column intensity is proportional to the $Z^{\sim2}$ ($Z$ is the atomic number)[37], thereby the bright and dark columns are the A-site (i.e., Pr, Ba, and Sr (green)) and the B-site (i.e., Fe (purple)), respectively, in the HAADF-STEM mode.

**Examination and characterization of exsolved particle size.** In general, particle size and surface distribution of catalysts have a considerable influence on the catalytic activity[4,38]. As such, the particle size and surface distribution of exsolved metal particles via reduction treatment could impact on the electro-catalytic activity of catalysts. Before examining the electro-catalytic effect of the in-situ exsolved Fe metal particles, an explicit comparison of exsolved particle size and surface distribution for R-PBSF00, R-PBSF30, and R-PBSF50 samples were presented in scanning electron microscope (SEM) images (Fig. 3c, f and Supplementary Figs. 9, 10). As shown in Fig. 3f, many small particles with size of about 100 to 200 nm are observed and uniformly socketed onto the perovskite oxide matrix after reduction, which are speculated as Fe metal particles. In contrast, the size of exsolved particles was relatively larger for R-PBSF00 and R-PBSF50 (Supplementary Figs. 11, 12). The energy dispersive spectroscopy (EDS) spectrum and the elemental mapping images of R-PBSF30 also clearly revealed that Fe metal particle with size of about 150 nm is well-socketed onto the R-P perovskite after reduction (Fig. 3g, h). Furthermore, noticeable energy shift to the higher energy in X-ray absorption near-edge structure (XANES), much increase in Fe-Fe shell intensity from the Fourier-transformed extended X-ray absorption fine structure (EXAFS) spectra after reduction, and the presence of $Fe^0$ $2p_{1/2}$ peak for only R-PBSF30 from X-ray photoelectron spectroscopy (XPS) measurements confirm the exsolution of Fe metal onto the surface under reducing condition[39], in coincidence with the above experimental results (Fig. 4). To investigate the electrically conductive properties of the exsolved Fe metal particles, the electrical conductivities as a function of temperature under reducing atmosphere were measured for PBSF (Supplementary Fig. 13). The A-PBSF30 displayed the highest electrical conductivity value compared to other PBSF in the reduction environment coupled with sufficiently high electrical conductivity in the air atmosphere (Supplementary Fig. 13b)[40], suggesting that the A-PBSF30 is the potential electrode material for S-SOC electrode material.

**Electrochemical performance evaluation.** Prior to assessment of electrochemical performance for A-PBSF30 in the practical application of S-SOCs, great thermo-chemical compatibility between all PBSF and the $La_{0.9}Sr_{0.1}Ga_{0.8}Mg_{0.2}O_{3-\delta}$ (LSGM) electrolyte was confirmed by XRD measurement (Supplementary Fig. 14). Moreover, similar microstructures of air-sintered PBSF samples imply that the electrochemical performance would not be affected by surface morphology (Supplementary Fig. 15). Then, electrochemical performance of symmetrical solid oxide fuel cells (S-SOFCs) using PBSF as both electrodes was characterized by LSGM electrolyte-supported cells to identify the huge impact of the exsolved Fe metal particle size and surface distribution (Fig. 5a and Supplementary Figs. 16, 17). The peak power density of the A-PBSF30 symmetrical cell is 1.23 W cm$^{-2}$ at 800 °C with humidified $H_2$ (3% $H_2O$) as fuel. This outstanding cell performance is the highest out of open literature based on LSGM electrolyte-supported S-SOFCs without any external catalysts at 800 °C under humidified $H_2$ (3% $H_2O$) as fuel to our best knowledge (Fig. 5b and Table 1)[28,39,41-46]. In addition, peak power output of 0.73 W cm$^{-2}$ was demonstrated in humidified $C_3H_8$ (3% $H_2O$) at 800 °C (Fig. 5c and Supplementary Fig. 18). Furthermore, the A-PBSF30 symmetrical cell demonstrated fairly stable current density without observable degradation for about 200 h in $H_2$ and 150 h in $C_3H_8$ at 700 °C (Fig. 5d, e). We also evaluated the electrochemical performance of the A-PBSF30 symmetrical cell in co-electrolysis mode. The excellent current density of −1.62 A cm$^{-2}$ at a cell voltage of 1.5 V (close to thermo-neutral voltage)[47] at 800 °C under co-electrolysis condition was demonstrated for the A-PBSF30 symmetrical cell (Fig. 5f), which is exceptionally high compared to other oxygen-conducting solid oxide electrolysis cell (SOEC) systems with different electrode materials[19,31,48,49]. The *in-operando* quantitative analysis of the synthetic gas products ($H_2$ and CO) was further investigated via gas chromatography (GC) profiles for the A-PBSF30 symmetrical cell at 800 °C during co-electrolysis of $H_2O$ and $CO_2$ (Supplementary Fig. 19). The amount of generated $H_2$ and CO were measured to be 0.50 and 10.81 ml min$^{-1}$ cm$^{-2}$, respectively, implying that the A-PBSF30 symmetrical cell could efficiently produce synthetic gas during co-electrolysis[50]. Together with superior electrochemical performance and efficient synthetic gas production, relatively constant voltage was observed for 100 h under a constant current load of −0.25 A cm$^{-2}$ at 700 °C in co-electrolysis mode (Fig. 5g), representing great durability in continuous SOEC operation. It is noteworthy that in-situ exsolution of well-dispersed Fe metal particles after complete phase reconstruction to R-P perovskite matrix acts as catalysts with promising electro-catalytic activity (Fig. 6), leading to outstanding electrochemical performances in various applications.

**Discussion**

In summary, this study successfully calculated $G_{vf-O}$ value from PrO and $TO_2$ in $Pr_{0.5}(Ba/Sr)_{0.5}TO_{3-\delta}$ (T = Mn, Fe, Co, and Ni) as the key factor for identifying the type of the phase reconstruction. Remarkably, the phase diagram acquired from in-situ

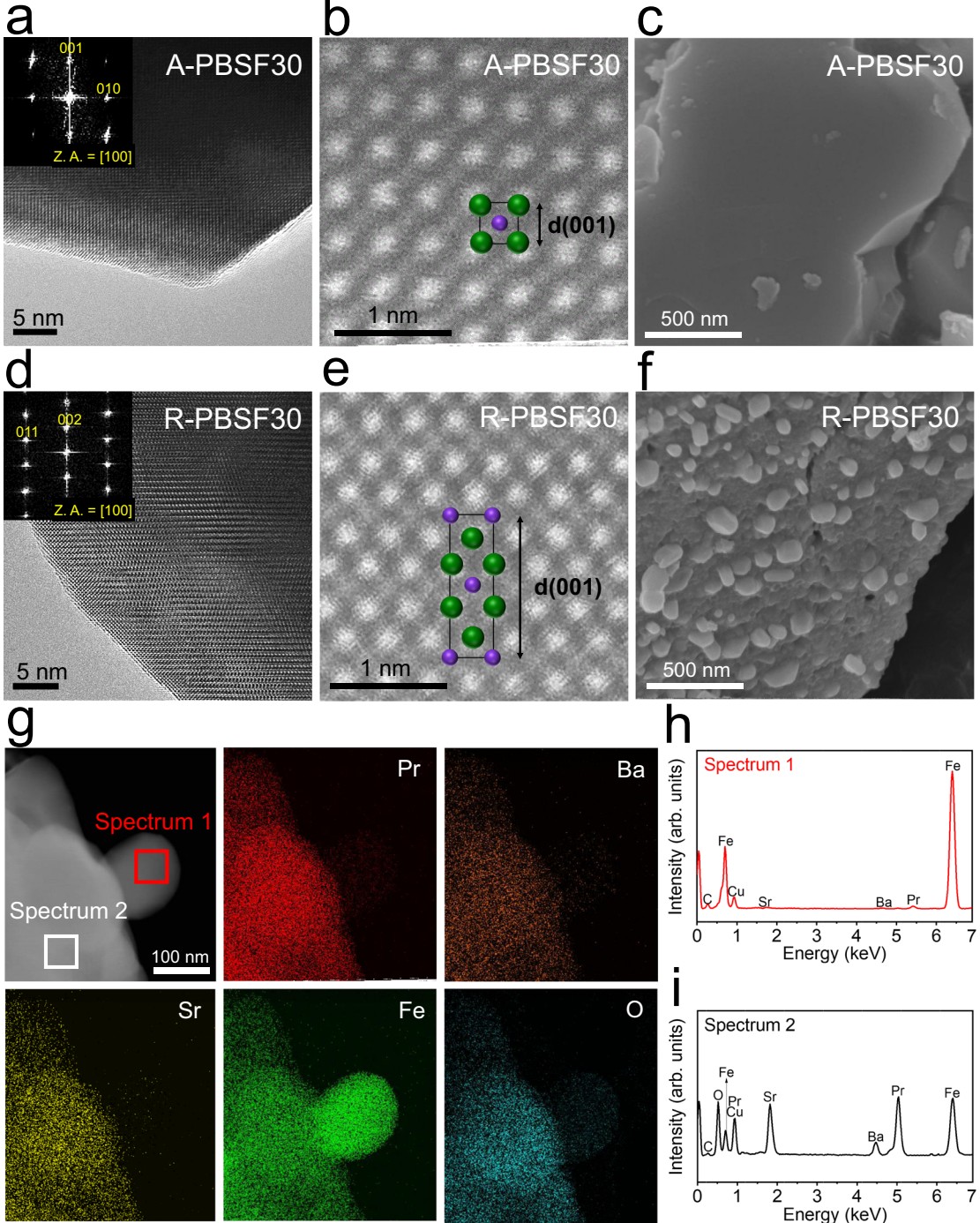

**Fig. 3 Electron microscopic analysis. a**, **b**, **d**, **e** Transmission electron microscopy (TEM) analysis. **a** High-resolution (HR) TEM image and the corresponding fast-Fourier transformed (FFT) pattern of $Pr_{0.5}Ba_{0.2}Sr_{0.3}FeO_{3-\delta}$ (A-PBSF30) with zone axis (Z.A.) = [100] and **b** high-angle annular dark-field (HAADF) scanning TEM (STEM) image of A-PBSF30 with simple perovskite structure of [100] direction with d-spacing 001. **d** HR TEM image and the corresponding FFT pattern of $(Pr_{0.5}Ba_{0.2}Sr_{0.3})_2FeO_{4+\delta}$ – Fe metal (R-PBSF30) with Z.A. = [100] and **e** HAADF STEM image and the atomic arrangement of R-PBSF30 of [100] direction with d-spacing 001. **c**, **f** Scanning electron microscope (SEM) images. SEM images presenting the surface morphologies of **c** A-PBSF30 sintered at 1200 °C for 4 h in air atmosphere and **f** R-PBSF30 reduced at 850 °C for 4 h in humidified $H_2$ environment (3% $H_2O$). **g–i** Scanning TEM-energy dispersive spectroscopy (EDS) analysis. **g** HAADF image of R-PBSF30 and elemental mapping of Pr, Ba, Sr, Fe, and O, respectively. **h**, **i** EDS spectra of **h** the exsolved Fe metal particle (Spectrum 1, red) and **i** the parent material $(Pr_{0.5}Ba_{0.2}Sr_{0.3})_2FeO_{4+\delta}$ (Spectrum 2, black).

temperature and environment-controlled XRD measurements indicated that the complete phase reconstruction to R-P perovskite occurs at least approximately $x = 0.3$ above at the reduction temperature of 850 °C for PBSF system. Among PBSF with complete phase reconstruction, the highly-populated Fe

metal particles socketed on R-PBSF30 attributed to excellent electrochemical performances under both fuel cell (1.23 W cm$^{-2}$ at 800 °C under $H_2$ fuel) and electrolysis cell ($-1.62$ A cm$^{-2}$ at 1.5 V and 800 °C under $CO_2$ and $H_2O$ fuels) modes coupled with great durability. Our investigations strongly provide a pathway to

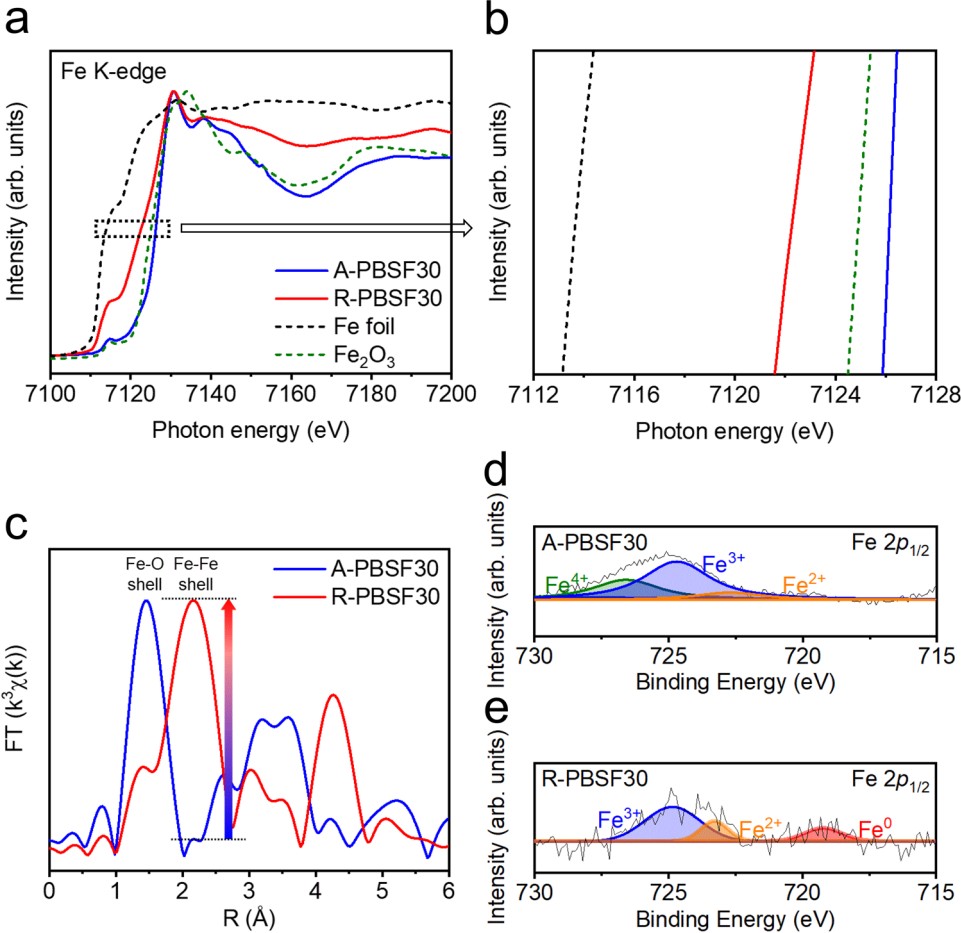

**Fig. 4 Oxidation state characterization. a, b** Fe K-edge X-ray absorption near-edge structure (XANES) spectra of $Pr_{0.5}Ba_{0.2}Sr_{0.3}FeO_{3-\delta}$ (A-PBSF30), $(Pr_{0.5}Ba_{0.2}Sr_{0.3})_2FeO_{4+\delta}$ (R-PBSF30) with two references (Fe foil and $Fe_2O_3$). **c** Fourier-transformed Fe K-edge extended X-ray absorption fine structure (EXAFS) spectra of A-PBSF30 and R-PBSF30. X-ray photoelectron spectra (XPS) of Fe $2p_{1/2}$ for **d** A-PBSF30 and **e** R-PBSF30.

explore new factors for the phase reconstruction and offer a new opportunity to discover prospective candidates with customized functionalities for next-generation energy-related applications.

## Methods

**Material synthesis**. $Pr_{0.5}Ba_{0.5-x}Sr_xFeO_{3-\delta}$ samples ($x = 0, 0.1, 0.2, 0.3, 0.4,$ and 0.5, abbreviated as PBSF in Supplementary Table 1) and $Pr_{0.5}Sr_{0.5}MnO_{3-\delta}$ were synthesized by the Pechini method. For PBSF materials, stoichiometric amounts of $Pr(NO_3)_3 \cdot 6H_2O$ (Aldrich, 99.9%, metal basis), $Ba(NO_3)_2$ (Aldrich, 99 + %), $Sr(NO_3)_2$ (Aldrich, 99+%) and $Fe(NO_3)_3 \cdot 9H_2O$ (Aldrich, 98+%) nitrate salts were dissolved in distilled water with the addition of quantitative amounts of citric acid and poly-ethylene glycol, while for $Pr_{0.5}Sr_{0.5}MnO_{3-\delta}$ material, stoichiometric amounts of $Pr(NO_3)_3 \cdot 6H_2O$ (Aldrich, 99.9%, metal basis), $Sr(NO_3)_2$ (Aldrich, 99+ %) and $Mn(NO_3)_2 \cdot 4H_2O$ (Aldrich, 97+%) nitrate salts were dissolved in distilled water with the addition of quantitative amounts of citric acid and poly-ethylene glycol. After removal of excess resin by heating at 280 °C, transparent organic resins containing metals in a solid solution were formed. The resins were calcined at 600 °C for 4 h and then sintered at 1200 °C for 4 h in air environment. The chemical compositions of the synthesized powders and their abbreviations are given in Supplementary Table 1.

**Structural characterization**. The crystal structures of the $Pr_{0.5}Ba_{0.5-x}Sr_xFeO_{3-\delta}$ samples ($x = 0, 0.3,$ and 0.5) and $Pr_{0.5}Sr_{0.5}MnO_{3-\delta}$ after heat-treated in two different environmental conditions (sintered at 1200 °C for 4 h in air environment and reduced at 850 °C for 4 h in humidified $H_2$ environment (3% $H_2O$)) were first identified by powder XRD patterns (Bruker diffractometer (LYNXEYE 1D detector), Cu $K\alpha$ radiation, 40 kV, 40 mA) in the 2 theta range of $20° < 2\theta < 60°$. To calculate the exact Bravais lattice of the PBSF, the samples were first pressed into pellets at 2 MPa for 30 s and then sintered at 1200 °C for 4 h in air atmosphere. The XRD patterns of air-sintered PBSF series and $(Pr_{0.5}Ba_{0.2}Sr_{0.3})_2FeO_{4+\delta}$ – Fe metal (R-PBSF30) samples were further measured by high-power (HP) XRD. (Max 2500 V, Cu $K\alpha$ radiation, 40 kV, 200 mA) at a scanning rate of $1° \min^{-1}$ and a range of $15° < 2\theta < 105°$. After the HP XRD

measurement, the powder patterns and lattice parameters were analyzed by the Rietveld refinement technique using the GSAS II program. The surface analysis of $Pr_{0.5}Ba_{0.2}Sr_{0.3}FeO_{3-\delta}$ (A-PBSF30) sintered at 1200 °C for 4 h in air atmosphere and $(Pr_{0.5}Ba_{0.2}Sr_{0.3})_2FeO_{4+\delta}$ – Fe metal (R-PBSF30) reduced at 850 °C for 4 h in humidified $H_2$ environment (3% $H_2O$) were conducted on XPS analyses on ESCALAB 250 XI from Thermo Fisher Scientific with a monochromated Al-$K\alpha$ (ultraviolet He1, He2) X-ray source. The X-ray absorption fine structure (XAFS) spectra of Fe K-edge for A-PBSF30, R-PBSF30, and two references (Fe foil and $Fe_2O_3$ powder) were measured on ionization detectors under fluorescence mode at the Pohang Accelerator Laboratory (PAL, 6D extended XAFS (EXAFS)). The XAFS and Fourier-transformed (FT) EXAFS spectra analysis were performed using the Athena (Demeter) program.

**In-situ phase reconstruction tendency evaluation**. The in-situ phase reconstruction tendency of $Pr_{0.5}Ba_{0.5-x}Sr_xFeO_{3-\delta}$ ($x = 0, 0.1, 0.2, 0.25, 0.3, 0.4,$ and 0.5) samples were identified by in-situ XRD measurements under humidified $H_2$ condition (3% $H_2O$). The $Pr_{0.5}Ba_{0.25}Sr_{0.25}FeO_{3-\delta}$ (A-PBSF25) sample was additionally synthesized by the Pechini method to evaluate the phase reconstruction tendency under reducing atmosphere. The $Pr_{0.5}Ba_{0.5-x}Sr_xFeO_{3-\delta}$ ($x = 0, 0.1, 0.2, 0.25, 0.3, 0.4,$ and 0.5) samples were sintered at 1200 °C for 4 h in air atmosphere to form simple perovskite structure with fine crystallinity. The reduction temperatures were ranged from 700 to 870 °C and 2 h were delayed at each temperature interval (Bruker D8 advance).

**Electron microscopy analysis**. The microstructures of (1) $Pr_{0.5}Ba_{0.5-x}Sr_xFeO_{3-\delta}$ samples ($x = 0, 0.3,$ and 0.5) sintered at 1200 °C for 4 h in air atmosphere, (2) $Pr_{0.5}Ba_{0.5-x}Sr_xFeO_{3-\delta}$ samples ($x = 0, 0.3,$ and 0.5) reduced at 850 °C for 4 h in humidified $H_2$ environment (3% $H_2O$), and (3) all PBSF samples sintered at 950 °C for 4 h in air atmosphere were investigated by using an SEM (Nova Nano FE-SEM). TEM analyses were conducted with a JEOL JEM 2100 F with a probe forming (STEM) Cs (spherical aberration) corrector at 200 kV.

**Electrical conductivity measurements**. The electrical conductivities of PBSF with respect to temperature were measured under air and 5% $H_2$ environments by the

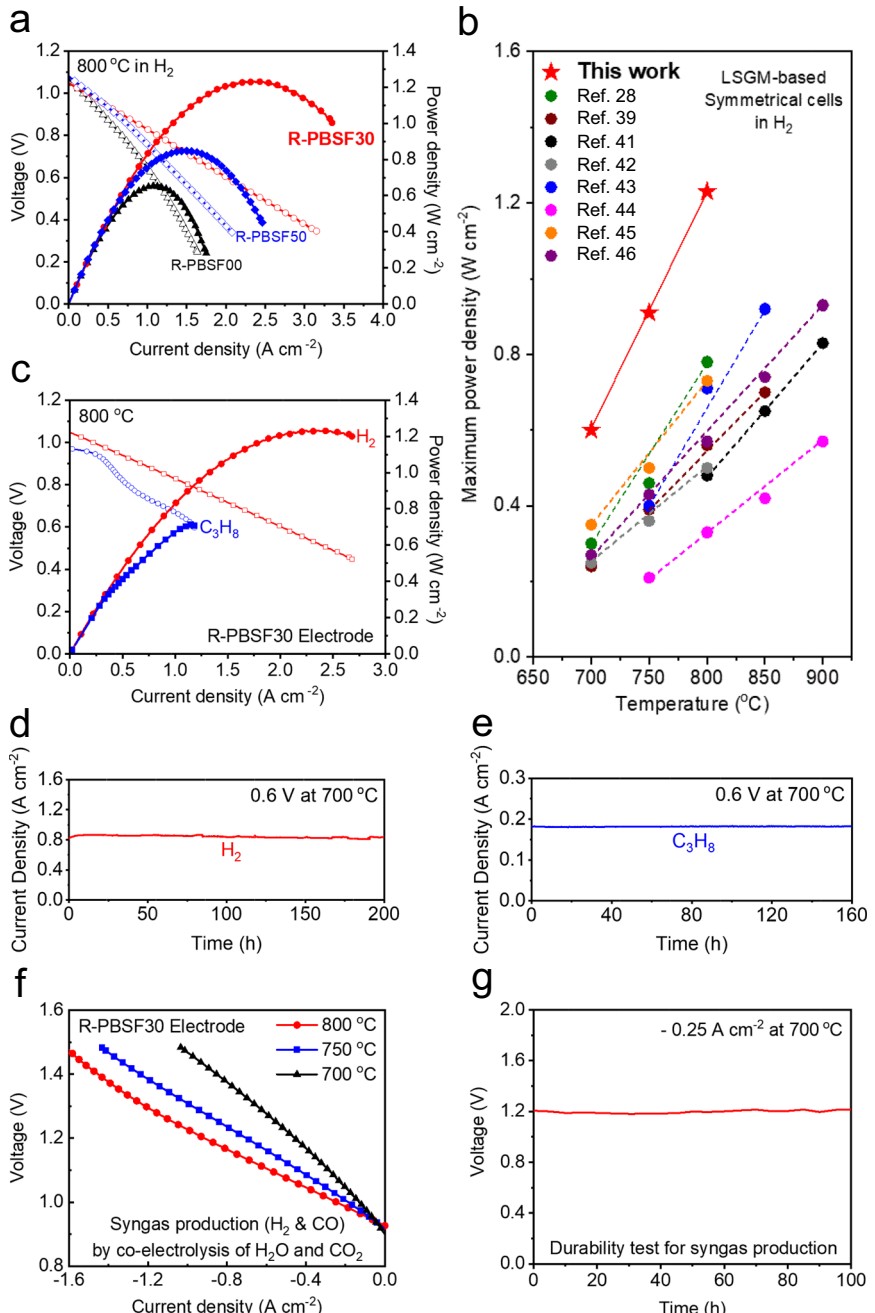

**Fig. 5 Electrochemical performance measurements. a, b** Comparison of the maximum power density values at 800 °C in $H_2$. (**a**) in terms of $Pr_{0.5}Ba_{0.5-x}Sr_xFeO_{3-\delta}$ compositions ($x = 0$, 0.3, and 0.5) and **b** of the present work and other LSGM electrolyte-supported studies with symmetrical cell configuration at various temperature regimes. **c** I–V curves and the corresponding power densities of symmetrical cell with $(Pr_{0.5}Ba_{0.2}Sr_{0.3})_2FeO_{4+\delta}$ – Fe metal (R-PBSF30) fuel electrode at 800 °C under $H_2$ and $C_3H_8$ humidified fuels (3% $H_2O$) fed on the fuel electrode and air fed on the air electrode. **d, e** Durability test of symmetrical cell with R-PBSF30 fuel electrode recorded with respect to time at a constant voltage of 0.6 V at 700 °C under **d** $H_2$ and **e** $C_3H_8$ humidified fuels. **f** I–V curves for symmetrical cell with R-PBSF30 fuel electrode with humidified $H_2$ and $CO_2$ with $H_2O$ co-fed to the fuel electrode side and air fed to the air electrode. **g** Durability test of symmetrical cell with R-PBSF30 fuel electrode recorded at a constant current of −0.25 A cm$^{-2}$ at 700 °C during co-electrolysis for 100 h.

4-probe method. The samples were pressed into pellets of cylindrical shape and then sintered at 1400 °C for 4 h in air environment to reach an apparent density of ~90%. The electrical conductivities were first measured in air atmosphere from 300 to 800 °C with intervals of 50 °C, and then measured in wet 5% $H_2$ atmosphere (Ar balance, 3% $H_2O$) from 300 to 800 °C with intervals of 50 °C. The current and voltage were recorded by a Biologic Potentiostat to calculate the resistance, resistivity, and conductivity of samples.

**Computational methods.** DFT calculations were performed to investigate the appropriate dopants for the phase reconstruction to n = 1 R-P perovskite along with the role of $Sr^{2+}$ concentration on phase reconstruction tendency of PBSF using the Vienna ab initio Simulation Package[51,52]. For the exchange-correlation, the generalized gradient approximation (GGA) based Predew-Burke-Ernzerhof functional was used[53]. The electron-ion interactions were described using the projector augmented wave potential[54,55]. A plane wave was expanded up to cutoff energy of 400 eV. Electronic occupancies were calculated using Gaussian smearing with a smearing parameter of 0.05 eV. For the bulk optimization, all internal atoms were relaxed using a conjugate gradient algorithm until the forces of each atom were lowered below 0.03 eV/Å with an energy convergence of $10^{-5}$ eV. GGA + U approach was used to correct the self-interaction errors with $U_{eff} = 4.0$ eV for Fe 3d orbital, $U_{eff} = 3.3$ eV for Co 3d orbital,

**Table 1 Comparison of the electrochemical performance of La$_{1-x}$Sr$_x$Ga$_{1-y}$Mg$_y$O$_{3-\delta}$ (LSGM) electrolyte-supported symmetrical solid oxide fuel cells (S-SOFCs) reported in the literature and in the present study.**

| Reference | Cell configuration (Air electrode \| Electrolyte \| Fuel electrode) | Electrolyte thickness (μm) | Maximum power density at 800 °C (W cm$^{-2}$) |
|---|---|---|---|
| Present study | A-PBSF30 \| LDC \| LSGM \| LDC \| R-PBSF30 | ~250 | 1.23 |
| (28) | Pr$_{0.4}$Sr$_{0.6}$Co$_{0.2}$Fe$_{0.7}$Nb$_{0.1}$O$_{3-\delta}$ \| LSGM \| Pr$_{0.8}$Sr$_{1.2}$(Co,Fe)$_{0.8}$Nb$_{0.2}$O$_{4+\delta}$ – CFA (Co-Fe alloy) | ~300 | 0.78 |
| (39) | Sr$_2$Ti$_{0.8}$Co$_{0.2}$FeO$_6$ \| LSGM \| Sr$_2$Ti$_{0.8}$Co$_{0.2}$FeO$_6$ | ~270 | 0.56 |
| (41) | Sr$_2$Fe$_{1.5}$Mo$_{0.5}$O$_{6-\delta}$ \| LSGM \| Sr$_2$Fe$_{1.5}$Mo$_{0.5}$O$_{6-\delta}$ | ~265 | 0.48 |
| (42) | Pr$_{0.6}$Sr$_{0.4}$Fe$_{0.7}$Ni$_{0.2}$Mo$_{0.1}$O$_{3-\delta}$ \| GDC \| LSGM \| GDC \| Pr$_{0.6}$Sr$_{0.4}$Fe$_{0.7}$Ni$_{0.2}$Mo$_{0.1}$O$_{3-\delta}$ | ~320 | 0.50 |
| | Pr$_{0.6}$Sr$_{0.4}$Fe$_{0.8}$Ni$_{0.2}$O$_{3-\delta}$ \| GDC \| LSGM \| GDC \| Pr$_{0.6}$Sr$_{0.4}$Fe$_{0.8}$Ni$_{0.2}$O$_{3-\delta}$ | ~320 | 0.44 |
| (43) | PrBa(Fe$_{0.8}$Sc$_{0.2}$)$_2$O$_{5+\delta}$ \| LSGM \| PrBa(Fe$_{0.8}$Sc$_{0.2}$)$_2$O$_{5+\delta}$ | ~275 | 0.71 |
| (44) | SmBaMn$_2$O$_{5+\delta}$ \| LSGM \| SmBaMn$_2$O$_{5+\delta}$ | ~300 | 0.33 |
| (45) | La$_{0.5}$Sr$_{0.5}$Fe$_{0.9}$Mo$_{0.1}$O$_{3-\delta}$ \| LSGM \| SDC \| La$_{0.5}$Sr$_{0.5}$Fe$_{0.9}$Mo$_{0.1}$O$_{3-\delta}$ | ~300 | 0.73 |
| (46) | La$_{0.6}$Sr$_{0.4}$Co$_{0.2}$Fe$_{0.7}$Mo$_{0.1}$O$_{3-\delta}$ \| SDC \| LSGM \| SDC \| La$_{0.6}$Sr$_{0.4}$Co$_{0.2}$Fe$_{0.7}$Mo$_{0.1}$O$_{3-\delta}$ | ~270 | 0.74 |

A-PBSF30: Pr$_{0.5}$Ba$_{0.2}$Sr$_{0.3}$FeO$_{3-\delta}$.
R-PBSF30: (Pr$_{0.5}$Ba$_{0.2}$Sr$_{0.3}$)$_2$FeO$_{4+\delta}$ – Fe metal.
LDC: La$_{0.4}$Ce$_{0.6}$O$_{2-\delta}$.
GDC: Gd$_{0.4}$Ce$_{0.6}$O$_{2-\delta}$.
SDC: Sm$_{0.4}$Ce$_{0.6}$O$_{2-\delta}$.
LSGM: La$_{1-x}$Sr$_x$Ga$_{1-y}$Mg$_y$O$_{3-\delta}$ ($x = 0.1$ or $0.2$ and $y = 0.15$ or $0.2$).

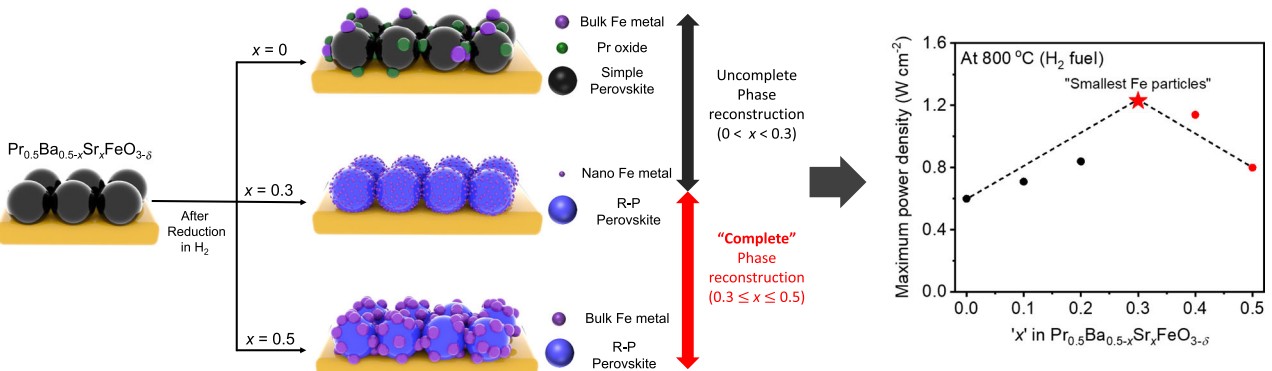

**Fig. 6 Schematic illustration of this work.** Schematic illustration of the fuel electrode side of Pr$_{0.5}$Ba$_{0.5-x}$Sr$_x$FeO$_{3-\delta}$ ($x = 0$, 0.3, and 0.5) symmetrical cells and its relation to electrochemical performances.

$U_{eff} = 4.0$ eV for Mn 3d orbital, $U_{eff} = 7.0$ eV for Ni 3d orbital, and $U_{eff} = 6.0$ eV for Pr 4f orbital[18,56,57]. For the Brillouin zones of the formation energy calculation of cubic perovskite ($2 \times 2 \times 4$ super cell) and n = 1 R-P perovskite ($2 \times 2 \times 1$ super cell), $3 \times 3 \times 1$ and $3 \times 3 \times 2$ Monkhorst-Pack k-point sampling were used, respectively[58]. For the oxygen vacancy formation energy calculations of BO$_2$ layer between two AO layers, PrO-terminated (001) slab model ($2 \times 2$ surface, 8 layers with 2 fixed bottom layers, vacuum layer of 16 Å) were used. For the co-segregation energy calculations, FeO$_2$-terminated (001) slab model ($2\sqrt{2} \times 2\sqrt{2}$ surface, 8 layers with 3 fixed bottom layers, vacuum layer of 16 Å) were used. For the Brillouin zones of the oxygen vacancy formation energy and the co-segregation energy calculations, $3 \times 3 \times 1$ and $1 \times 1 \times 1$ Monkhorst-Pack k-point sampling were used. The optimized lattice parameters of four materials, A-PBSF00 (Ba:Sr = 16:0), Pr$_{0.5}$Ba$_{0.1875}$Sr$_{0.3125}$FeO$_{3-\delta}$ (Ba:Sr = 6:10), Pr$_{0.5}$Ba$_{0.125}$Sr$_{0.375}$FeO$_{3-\delta}$ (Ba:Sr = 4:12), and A-PBSF50 (Ba:Sr = 0:16) were used for model structures in the computational studies. For the Ba$^{2+}$/Sr$^{2+}$ mixed models, the most stable configurations among the total 5 different Ba configurations were used.

The relative energies required for the phase reconstruction from simple perovskite to n = 1 R-P perovskite (E$_{recon}$) of six model structures with different Sr$^{2+}$ concentration were calculated using the total energy difference between simple perovskite and n = 1 R-P perovskite by following equation:

$$E_{recon} = \frac{1}{16}E_{R-Pperov} + \frac{1}{2}E_{Fe} + \frac{1}{2}E_{O_2} - \frac{1}{16}E_{simpleperov}, \quad (3)$$

Where $E_{R-P\ perov}$ and $E_{simple\ perov}$ are the total energy of simple perovskite ($2 \times 2 \times 4$ super cell) and R-P perovskite ($2 \times 2 \times 1$ super cell), $E_{Fe}$ is total energy of body-centered cubic Fe metal unit cell, and $E_{O_2}$ is the total energy of gas phase oxygen molecule.

The oxygen vacancy formation energies (E$_{vf-O}$) of Pr$_{0.5}$Ba$_{0.5}$TO$_{3-\delta}$, Pr$_{0.5}$Sr$_{0.5}$TO$_{3-\delta}$ (T = Mn, Co, Fe, and Ni), and four model structures with different Sr$^{2+}$ concentrations were calculated using the lattice oxygen on the BO$_2$ layer since the phase reconstruction from simple perovskite (ABO$_3$) to n = 1 R-P perovskite

(A$_2$BO$_4$) requires the formation of both oxygen and B-site vacancies. For the Pr$_{0.5}$Ba$_{0.5}$TO$_{3-\delta}$ and Pr$_{0.5}$Sr$_{0.5}$TO$_{3-\delta}$ (T = Mn, Co, Fe, and Ni) models, the most stable structure configurations were utilized for the oxygen vacancy formation energy calculations. For the four computational models with different Sr$^{2+}$ concentrations, the most stable vacancy sites were utilized for Ba$^{2+}$/Sr$^{2+}$ mixed models with Ba:Sr = 6:10 and Ba:Sr = 4:12. The E$_{vf-O}$ was calculated by following equation:

$$E_{vf-O} = E_{perov-defect} + \frac{1}{2}E_{O_2} - E_{perov} \quad (4)$$

where $E_{perov-defect}$ and $E_{perov}$ are the total energies of PrO-terminated (001) perovskite slab model with and without the oxygen vacancy, respectively.

The co-segregation energy (E$_{co-seg}$) is defined as the total energy difference of two surface models that have different vacancy site. The co-segregation energies of four computational models with different Sr$^{2+}$ concentrations were calculated by following equation:

$$E_{co-seg} = E_{(Fe-V_O)\ surface} - E_{(Fe-V_O)\ bulk} \quad (5)$$

where $E_{(Fe-V_O)\_surface}$ and $E_{(Fe-V_O)\_bulk}$ are total energies of FeO$_2$-terminated (001) perovskite slab model that have oxygen vacancy on surface FeO$_2$ and bulk FeO$_2$ layer, respectively.

Furthermore, the Gibbs free energy for oxygen vacancy formation of eight samples were calculated to include the temperature and oxygen partial pressure factors in the E$_{vf-O}$ calculations. The equations used for the G$_{vf-O}$ calculations from the surface AO and BO$_2$ layers in Pr$_{0.5}$(Ba/Sr)$_{0.5}$TO$_{3-\delta}$ (T = Mn, Fe, Co, and Ni) are as follows:

$$G_{vf-O}(AOlayer) = E_{perov-defect} - E_{perov} + \frac{1}{2}\mu_{O_2} \quad (6)$$

$$\mu_{O_2} = E^{DFT}_{O_2(g)} + E^{ZPE}_{O_2(g)} + E^{correction}_{O_2(g)} - TS_{O_2(g)} + k_B T\ln\left(\frac{P_{O_2}}{P_0}\right) \tag{7}$$

$$G_{vf-O}(BO_2 \text{layer}) = E_{perov-defect} - E_{perov} + \left(\mu_{H_2O} - \mu_{H_2}\right) + E^{O_{vac}diffusion}_a \tag{8}$$

$$\mu_{H_2O} = \left(\triangle H^{exp}_{H_2O} + E^{DFT}_{H_2(g)} + E^{ZPE}_{H_2(g)} + \frac{1}{2}\left(E^{DFT}_{O_2(g)} + E^{ZPE}_{O_2(g)} + E^{correction}_{O_2(g)}\right)\right) \\ - TS_{H_2O(g)} + k_B T\ln\left(\frac{P_{H_2O}}{P_0}\right) \tag{9}$$

$$\mu_{H_2} = E^{DFT}_{H_2(g)} + E^{ZPE}_{H_2(g)} - TS_{H_2(g)} + k_B T\ln\left(\frac{P_{H_2}}{P_0}\right) \tag{10}$$

The $E_{perov-defect}$ and $E_{perov}$ are the total energies of PrO-terminated (001) perovskite slab model with and without the oxygen vacancy, respectively. The $\mu_{O_2}$, $\mu_{H_2}$, and $\mu_{H_2O}$ are the Gibbs free energy of di-atomic oxygen molecule, hydrogen, and water molecule, respectively. The $E^{DFT}_{O_2(g)}$ and $E^{DFT}_{H_2(g)}$ is the gas phase energy of ground state triplet $O_2$ molecule and hydrogen molecule, respectively. The zero point energies of oxygen molecule ($E^{ZPE}_{O_2(g)}$) and hydrogen molecule ($E^{ZPE}_{H_2(g)}$) were extracted from the previous calculated value[59]. The standard entropy of gas phase oxygen ($S_{O_2(g)}$) was obtained from National Institute of Standards and Technology Chemistry Web-Book (http://webbook.nist.gov/chemistry). Moreover, the correction energy of oxygen molecule ($E^{correction}_{O_2(g)}$) was added to reconcile the $E_{vf-O}$ differences between the results achieved via computational method (GGA functional) and real experimental results[60]. The temperature and p($O_2$) are 750ºC and $10^{-9}$ atm for $G_{vf-O}$ calculations at the surface AO layer (Eq. 6 and 7) and the temperature, p($O_2$), p($H_2$), and p($H_2O$) values are 750ºC, $10^{-9}$, 0.1, and 0.01 atm, respectively, for $G_{vf-O}$ calculations at the $BO_2$ layer (Eq. 8, 9, and 10). Under this specified condition (reducing condition), we assumed that the reduction of $BO_2$ layer occurred via two elementary steps: surface hydrogen oxidation reaction ($O_{lattice} + H_2(g) \leftrightarrow H_2O(g)$) and oxygen vacancy diffusion toward the $BO_2$ layer. The activation energy of oxygen vacancy diffusion ($E^{O_{vac}diffusion}_a$: 0.95 eV) was calculated from the electrochemical measurements (Arrhenius plot of area specific resistance) of $Pr_{0.4}Sr_{0.6}Fe_{0.875}Mo_{0.125}O_{3-\delta}$ (PSFM) material under $H_2$ condition[61].

**Electrochemical performance measurements**. $La_{0.9}Sr_{0.1}Ga_{0.8}Mg_{0.2}O_{3-\delta}$ (LSGM) powder was prepared by conventional solid-state reaction to fabricate LSGM electrolyte-supported symmetrical S-SOCs. Stoichiometric amounts of $La_2O_3$ (Sigma, 99.99%), $SrCO_3$ (Sigma, 99.99%), $Ga_2O_3$ (Sigma, 99.99%) and MgO (Sigma, 99.9%) powders were first mixed in a mortar and then ball-milled in ethanol for 24 h to obtain the desired composition. After drying, the obtained powder was calcined at 1000 °C for 6 h. After formation of LSGM powder with desired stoichiometry, the electrolyte substrate was prepared by pressing at 2 MPa for 30 s into cylindrical shape and then sintered at 1475 °C for 5 h. The thickness of LSGM electrolyte was polished to about 250 μm. A $La_{0.6}Ce_{0.4}O_{2-\delta}$ (LDC) as a buffer layer was also prepared by ball-milling stoichiometric amounts of $La_2O_3$ (Sigma, 99.99%) and $CeO_2$ (Sigma, 99.99%) in ethanol for 24 h and then calcined at 1000 °C for 6 h. Electrode slurries were prepared by mixing pre-calcined powders of PBSF with an organic binder (Heraeus V006) in 3:6:0.6 weight ratio. The electrode inks were applied onto the LSGM pellet by a screen-printing method and then sintered at 950 °C for 4 h in air to achieve the desired porosity. The porous electrodes had an active area of 0.36 cm$^2$ and a thickness of about 15 μm. The LDC layer was screen-printed between the electrode and electrolyte to prevent inter-diffusion of ionic species between electrode and electrolyte. The cells with configuration of Electrode│LDC│LSGM│LDC│Electrode were mounted on alumina tubes with ceramic adhesives (Ceramabond 552, Aremco) for electrochemical performance tests (Cross-sectional SEM image of the A-PBSF30 symmetrical cell given in Supplementary Fig. 20). Silver paste and silver wire were utilized for electrical connections to both the fuel electrode and air electrode. The entire cell was placed inside a furnace and heated to the desired temperature. I–V polarization curves of synthesized fuel cells with PBSF as both sides of electrodes were measured using a BioLogic Potentiostat in a temperature range of 700 to 800 °C (temperature interval: 50 °C) in humidified hydrogen (3% $H_2O$) at a flow rate of 100 ml min$^{-1}$. Fuel cell evaluation under humidified $C_3H_8$ fuel (3% $H_2O$) at a flow rate of 100 ml min$^{-1}$ were also performed for symmetrical solid oxide fuel cell (S-SOFC) test with cell composition A-PBSF30│LDC│LSGM│LDC│A-PBSF30 from 700 to 800 °C (temperature interval: 50 °C) using a BioLogic Potentiostat. For the electrochemical performance test of S-SOC with the cell composition A-PBSF30│LDC│LSGM│LDC│A-PBSF30 during co-electrolysis, 50 ml min$^{-1}$ of $H_2$ and $CO_2$ into a $H_2O$-containing bubbler and acetone (a heating tape) were co-fed to fuel electrode and 100 ml min$^{-1}$ of air was fed to air electrode. The in-operando quantitative analysis of the generated synthetic gas ($H_2$ and CO) during co-electrolysis of $CO_2$ and $H_2O$ (Ratio of $CO_2$:$H_2$:$H_2O$ = 45:45:10) for the A-PBSF30 symmetrical cell (@800 °C and 1.5 V) was demonstrated by the gas chromatograph (Agilent 7820 A GC instrument) with a thermal conductivity detector and a packed column (Agilent carboxen 1000).

**Data availability**
The data measured, simulated, and analyzed in this study are available from the corresponding author on reasonable request.

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

## Acknowledgements

This work was supported by the Korea Institute of Energy Technology Evaluation and Planning (KETEP) and the Ministry of Trade, Industry & Energy (MOTIE) of the Republic of Korea (No. 20213030030150) and the National Research Foundation (NRF) funded by the Ministry of Education (NRF-2019R1C1C1005801, NRF-2021M3I3A1084292, and NRF-2021R1A2C3004019). This work was also supported by "$CO_2$ utilization battery for hydrogen production based on fault-tolerance deep learning" (No. 1.200097.01). The Xray absorption fine structure experiments performed at the beamline 6D of Pohang Accelerator Laboratory was supported by the Pohang University of Science and Technology (POSTECH) and Ulsan National Institute of Science and Technology Central Research Facilities center (UCRF).

## Author contributions

H.K. and O.K. carried out most of the experimental works and contributed to manuscript writing. C.L. performed DFT calculations. M.T.C. gave help on additional DFT calculations. J.O. performed the gas chromatography (GC) analysis. H.Y.J. conducted TEM measurements and analyzed the TEM images. S.C., J.W.H., and G.K. designed the experiments and analyzed the data. All authors contributed to the discussions and analysis of the results regarding the manuscript.

## Competing interests

The authors declare no competing interests.
