## [Peer review file · Nature Communications]

REVIEWER COMMENTS

Reviewer #1 (Remarks to the Author):

In this work, an R-P perovskite electrode with exsolved Fe nanoparticles is developed through the reduction of an ABO₃ perovskite. Special attention is paid on the phase reconstruction process with in-situ XRD and DFT calculation. A symmetric cell with that electrode shows promising performance and stability as H₂/C₃H₈ SOFC and H₂O/CO₂ SOEC. However, some results and conclusions require more discussion and explanation. It could be accepted after a major revision.

Detailed comments:

1. As the authors summarized in the introduction, there is a large amount of closely relevant literature on the reconstruction of the perovskite to the R-P phase with the exsolution of metallic nanoparticles, and the authors said that the main objective of this work is the "comprehensive understanding of key factors modulating the phase reconstruction". As shown in Fig. 1a, the authors associated the phase reconstruction process only with the oxygen vacancy formation energy, and the R-P phase will be formed when the oxygen vacancy formation energy is in a special range. However, we know that the formation of oxygen vacancy and the phase transition are highly influenced by many factors such as temperature and oxygen partial pressure. So please provide more explanation on Fig. 1a. How to get those results? Is that a general conclusion, or only appropriate in some temperature and oxygen partial pressure ranges?

2. In Fig. 2b, the authors provided a comprehensive phase diagram based on in-situ XRD results. Take $x=0.5$ as an example. Based on Fig. 1a, it should be a R-P perovskite phase. However, as shown in Fig. 2b, it is a simple perovskite (region I) after the reduction below 770 oC, simple perovskite + R-P perovskite + Fe metal (region III) in 780-830 oC, and R-P perovskite + Fe metal (region IV) above 840 oC. So I wonder if the points represent equilibrium and stable states (which should be a basic requirement for a phase diagram), or just transient states due to the short reduction time? If the Pr_{0.5}Sr_{0.5}FeO₃ is reduced at 830 oC for a much longer time, can it transform to R-P perovskite + Fe metal (region IV)? If it is reduced at 850 oC for a much longer time, will the R-P perovskite decompose further?

3. In Fig. 2d-e, how to determine the concentration of oxygen vacancy (δ) in the material, which is remarkably influence by the temperature and oxygen partial pressure, before the DFT calculation?

4. Please provide the cross-sectional SEM image of the symmetric cell for the better understanding of the structure of the electrode layer, LDC interlayer, and their interface.

5. Why is the P_{max} of the symmetric cell with PBSF50 electrodes lower than that of the symmetric cell with PBSF30 electrodes? Is it due to the lower catalytic activity of PBSF50 for the electrochemical oxidation of H₂, or for the electrochemical reduction of O₂, or both? The authors only provided very limited I-V results of the symmetric cell in SOFC and SOEC modes, from which we can hardly separate the activities of the anode and the cathode from each other. For the better evaluation of the material, at least the EIS of the symmetric cells in H₂ and O₂ should be provided to reveal its exact activities towards H₂ oxidation and O₂ reduction. The performance of the electrode should be compared with other best anode and cathode materials, not only the electrodes for symmetric cells.

6. For the co-electrolysis of H₂O/CO₂, the quantitative analysis of the product is necessary. How much H₂ and CO are in the syngas?

Reviewer #2 (Remarks to the Author):

A comprehensive understanding of key parameters affecting the phase reconstruction from single perovskite to R-P perovskite is an interesting topic since most perovskites show phase reconstruction during reduction. But the main conclusion cannot be supported by the experimental results. This work is not suitable to be published in its present form in this journal. The questionable points are listed below.

1. The author declares that oxygen vacancy formation energies ($E_{\text{f-O}}$) from PrO and TO_2 in $\text{Pr}_{0.5}(\text{Ba/Sr})_{0.5}\text{TO}_3$ -d (T = Mn, Fe, Co, and Ni) are proposed as the important factor in determining the type of phase reconstruction in perovskites. However, this conclusion is only supported by T = Fe, how about the Mn for instance? The experiments on sample of T = Mn should be added.
2. The Sr doping can decrease the E_{recon} , but the difference between $x = 0$ and $x = 0.5$ are only 0.3 eV, it is hard to understand that such a small difference can impose a big influence to phase reconstruction. The author should also calculate the formation energy of RP phase with different Sr contents.
3. The assignment of Fe in X-ray photoelectron spectra is questionable. The noted binding energy value of Fe is too high.
4. XRD refinement should give the phase ratios of RP to Fe metal to confirm the equation (2).
5. In Supplementary Figure 2b, the XRD peak at 33° of R-PBSF3 looks much higher than the standard peak, why? And it is also not well refined in the XRD refinement (Supplementary Figure 3)
6. The conductivity of the RP phase is much low and the highest value is only 2 S cm^{-1} . Why could such high cell performance be obtained?
7. Some typos: in the part of Transmission electron microscopy analysis, "0.395 nm (Figure 3a) and 0.634 nm (Figure 3c)" and "the simple perovskite (Figure 3b) and R-P perovskite (Figure 3d)" should be changed to "0.395 nm (Figure 3a) and 0.634 nm (Figure 3d)" and "the simple perovskite (Figure 3b) and R-P perovskite (Figure 3e)".

Reviewer #3 (Remarks to the Author):

This manuscript deals with studying the oxygen vacancy formation energies in perovskites as a potential descriptor for predicting the amount of exsolved metallic particles upon strong reduction of the material. To achieve as much exsolved metal particles as possible, the authors claim that complete phase reconstruction from simple to Ruddlesden-Popper perovskite is the most suitable pathway. Their approach is a combination of DFT calculations (to predict the most suitable composition for complete phase reconstruction) and in-situ X-ray diffraction (to map the phase diagram, where the observed phase evolution is plotted as a function of A-site doping). Furthermore, the material with the most promising exsolution behaviour is tested for its suitability as an SOFC electrode. The DFT and in-situ XRD part are sound and especially the obtained phase diagram is a strong result. However, what I completely missed is a discussion on why the oxygen vacancy formation enthalpy determines the phase transition behaviour. In this question, the reader is left entirely to his or her own thoughts, which should not be the case with a paper in Nature Communications. There definitely needs to be more detailed explanation and discussion before the paper can be recommended for publication. Moreover, I unfortunately have to say that the electrochemical part of the paper is rather poor and needs major revisions. I am sorry for this harsh judgement, since I principally like the approach of the authors, but there are yet too much flaws in the manuscript and I therefore recommend that it is reconsidered in a largely revised version. Detailed comments and criticism: • The definition of the enthalpy limits between the different phase regions given in Fig.1 is crucial for a correct prediction of exsolution behaviour, therefore it needs more discussion. Where do the values given as limits come from? This appears to be especially important, as three or four of the calculated values are quite close at these limits. Hence, slight shifts in the defined limits or small errors from the calculation may lead to wrong predictions. I

suggest to compare the obtained oxygen vacancy formation enthalpies with several literature data. There is already quite some comparable data available - e.g. on (La,Sr)CoO₃ or (La,Sr)MnO₃ based materials. Moreover, quantitatively knowing the errors/uncertainties of the calculation results would be advantageous to being able judging the reliability of the calculated results. I am aware, that 2 commonly no errors are given for DFT results, since very often an accuracy of some 0.1eV is sufficient for supporting or disproving a hypothesis, which was gained by interpretation of experimental results. Here, however, the accuracy of the oxygen vacancy formation energies appears to be very important, since the DFT results are very close at the borderline that decides about which behaviour to expect. • Owing to my opinion several of the Figures from the supporting info should be moved to the main text. As far as I remember correctly, Nature Communications allows up to 7 display items. So there should be plenty of space available. • I do not agree with the fitting approach of the Fe XPS spectra. First of all, measuring only the Fe2p_{1/2} line provides not sufficient information for deciding the oxidation state of iron, since also the shake-up features between Fe2p_{1/2} and Fe2p_{3/2} peak need to be considered to draw such conclusions.[1] Second, identification of Fe⁴⁺ only from XPS is daring – in perovskites for example, localisation of electron holes at the Fe-O molecular orbital is more likely as already confirmed by XAS experiments. [2] • The electrical conductivity of an electrode material alone is not a suitable descriptor for predicting a good electrochemical behaviour. For a sufficiently thick 3D porous electrode with an electronic conductivity higher than the ionic conductivity – which appears to be the case for this material when looking at the relatively high values of total conductivity in Supp.Fig. 11 – the ionic conductivity and the surface reaction resistance are equally important, since the decay length of the electrochemical activity is $\sqrt{(R_{\text{react}}/R_{\text{ion}})}$. [3] • The chosen method is largely unsuitable for testing the electrochemical performance of the electrode material. The reason for this is that the performance of the electrolyte-supported SOFCs manufactured for this purpose is practically only limited by the ohmic resistance of the electrolyte in the investigated temperature range of 700-800°C. This can also be seen very clearly in the practically linear I-V characteristics (Figs. 4 and S14) at low to moderate current densities (the non-linearity at high current density points towards a concentration limitation). Especially at very low currents, the I-V-curves hardly show any discernible non-linearity, which could indicate an electrode polarisation limitation. This means that the high performance is mainly achieved either by a very good ion-conducting or comparatively thin LSGM electrolyte and therefore no quantitative statement about the electrode performance is permissible. Instead of cell measurements, a characterisation of the polarisation resistance of the electrodes would be much better suited to compare the investigated material with similar materials in the literature. However, it also needs to be noted that the polarisation resistance of porous electrodes strongly depends on their 3D structure (tortuosity, inner surface, etc.). This effect was also completely neglected in the comparison in Fig. 4. • Why did the authors decide not to use a Hubbard-U to properly consider effects of electron localisation? Especially for Fe-based perovskites this effect can be important (and at least in the XPS fits the authors do consider localised electrons). 3 References 1 M. Descostes, F. Mercier, N. Thomat, C. Beaucaire, M. Gautier-Soyer. Use of XPS in the determination of chemical environment and oxidation state of iron and sulfur samples: constitution of a data basis in binding energies for Fe and S reference compounds and applications to the evidence of surface species of an oxidized pyrite in a carbonate medium. *Appl. Surf. Sci.* 165 (2000), p.288-302 ([http://dx.doi.org/10.1016/S0169-4332\(00\)00443-8](http://dx.doi.org/10.1016/S0169-4332(00)00443-8)). 2 D.N. Mueller, M.L. Machala, H. Bluhm, W.C. Chueh. Redox Activity of Surface Oxygen Anions in Oxygen-Deficient Perovskite Oxides During Electrochemical Reactions. *Nat. Comm.* 6 (2015), (<https://dx.doi.org/10.1038/ncomms7097>). 3 S.B. Adler, J.A. Lane, B.C.H. Steele. Electrode Kinetics of

Porous Mixed-Conducting Oxygen Electrodes. *J. Electrochem. Soc.* 143 (1996), p.3554-3564
(<https://doi.org/10.1149/1.1837252>).

Author's Answers to Reviewers

The authors sincerely appreciate the editor and reviewers for providing valuable comments and suggestions that improves the quality of this manuscript. Each comment has been carefully considered point-by-point and responded. The responses to the comments are addressed as follows, and the corrections are highlighted with red color accordingly in the revised manuscript.

Answers to Reviewer #1

General Comments: In this work, an R-P perovskite electrode with exsolved Fe nanoparticles is developed through the reduction of an ABO_3 perovskite. Special attention is paid on the phase reconstruction process with in-situ XRD and DFT calculation. A symmetric cell with that electrode shows promising performance and stability as H_2/C_3H_8 SOFC and H_2O/CO_2 SOEC. However, some results and conclusions require more discussion and explanation. It could be accepted after a major revision.
Response to General Comments: We sincerely appreciate the reviewer for providing us insightful comments to improve the quality of this manuscript. We reflected the reviewer's comment as much as possible and provided detailed point-by-point responses for each comments below.
Detailed Comments: Comment 1. As the author summarized in the introduction, there is a large amount of closely relevant literature on the reconstruction of the perovskite to the R-P phase with the exsolution of metallic nanoparticles, and the authors said that the main objective of this work is the "comprehensive understanding of key factors modulating the phase reconstruction". As shown in Fig. 1a, the authors associated the phase reconstruction process only with the oxygen vacancy formation energy, and the R-P phase will be formed when the oxygen vacancy formation energy is in a special range. However, we know that the formation of oxygen vacancy and the phase transition are highly influenced by many factors such as temperature and oxygen partial pressure. So please provide more explanation on Fig. 1a. How to get those results? Is that a general conclusion, or only appropriate in some temperature and oxygen partial pressure ranges?
Response to Comment 1: We thank the reviewer for the valuable comment. We completely agree with the reviewer's comment that the formation of oxygen vacancies and the phase transition are highly influenced by many factors such as temperature and oxygen partial pressure. Accordingly, as the reviewer pointed out, we specified the detailed condition of the oxygen vacancy formation energy (E_{vf-O}) calculations with respect to the real experimental conditions for confirming the E_{vf-O} as one of the main factors modulating the phase reconstruction. In this manuscript, we initially compared the E_{vf-O} from the AO and BO_2 layers in $Pr_{0.5}(Ba/Sr)_{0.5}TO_{3-\delta}$ ($T = Mn, Fe, Co, \text{ and } Ni$) and proposed three possible ranges by two criteria (phase decomposition and phase reconstruction occurring under reducing condition): (i) Phase decomposition occurs under reducing condition when the A-site (AO layer) $E_{vf-O} < 1.5$ eV, (ii) Phase reconstruction to Ruddlesden-Popper (R-P) perovskite occurs under reducing condition when the A-site $E_{vf-O} > 1.5$ eV and 1.6 eV $<$ the B-site (BO_2 layer) $E_{vf-O} < 2.8$ eV, and (iii) Phase reconstruction to layered perovskite and/or the phase remains as simple perovskite when the A-site $E_{vf-O} > 1.5$ eV and the B-site $E_{vf-O} > 2.8$ eV. However, after careful

consideration, we thought that the calculated E_{vf-O} values could not provide the absolute standard for the phase decomposition and/or phase reconstruction (**Fig. 1a** in the original manuscript) because these values were relatively identified values that did not contain temperature and oxygen partial pressure factors. To contain the temperature and oxygen partial pressure terms in the E_{vf-O} calculations, we additionally calculated the Gibbs free energy difference oxygen vacancy formation (G_{vf-O}) by using the Gibbs free energies of oxygen, hydrogen, and water molecules. For reference, the abbreviations for eight samples used for the G_{vf-O} calculations are given in **Table R1** to remove confusion concerning sample information. The re-calculated G_{vf-O} for eight samples from the surface AO layer (green bar) and BO_2 layer (purple bar) are displayed in revised **Fig. 1a** (in the response letter). Unlike the E_{vf-O} calculations, since the G_{vf-O} contains temperature and oxygen partial pressure terms, we changed the standard value (borderline) for the phase reconstruction behavior: (i) Phase decomposition occurs under reducing condition when the A-site (surface AO layer) $G_{vf-O} < 0$ eV, (ii) Phase reconstruction to Ruddlesden-Popper (R-P) perovskite occurs under reducing condition when the A-site $G_{vf-O} > 0$ eV and -1.2 eV $<$ the B-site (BO_2 layer) $G_{vf-O} < 0$ eV, and (iii) Phase reconstruction to layered perovskite and/or the phase remains as simple perovskite when the A-site $G_{vf-O} > 0$ eV and the B-site $G_{vf-O} > 0$ eV.

The detailed explanation for the G_{vf-O} calculations from the surface AO and BO_2 layers in $Pr_{0.5}(Ba/Sr)_{0.5}TO_{3-\delta}$ (T = Mn, Fe, Co, and Ni) are as follows:

(1) Phase decomposition at 750 °C and $p(O_2) : 10^{-9}$ atm (Surface AO layer)

The phase for $Pr_{0.5}(Ba/Sr)_{0.5}CoO_{3-\delta}$ -based electrodes are reported to be decomposed at 700 °C under $p(O_2) < 10^{-6}$ atm ($p(O_2)$ similar to argon gas)^{R1}. Considering this experimental data, we re-calculated the G_{vf-O} values from the surface AO layer of $Pr_{0.5}(Ba/Sr)_{0.5}TO_{3-\delta}$ (T = Mn, Fe, Co, and Ni) at 750 °C and $p(O_2) : 10^{-9}$ atm (reducing condition) by using the following equations^{R2} (abbreviations given in **Table R1**):

Equation R1 and Equation R2:

$$G_{vf-O} = E_{perov-defect} - E_{perov} + \frac{1}{2}\mu_{O_2} \quad (R1)$$

$$\mu_{O_2} = E_{O_2(g)}^{DFT} + E_{O_2(g)}^{ZPE} + E_{O_2(g)}^{correction} - TS_{O_2(g)} + k_B T \ln\left(\frac{p_{O_2}}{P_0}\right) \quad (R2)$$

The $E_{perov-defect}$ and E_{perov} are the total energies of PrO-terminated (001) perovskite slab model with and without the oxygen vacancy, respectively. The μ_{O_2} is the Gibbs free energy of the di-atomic oxygen molecule. The $E_{O_2(g)}^{DFT}$ is the gas phase energy of ground state triplet O_2 molecule. The zero point energy ($E_{O_2(g)}^{ZPE}$) and was extracted from the previous calculated value^{R3}. The standard entropy of gas phase oxygen ($S_{O_2(g)}$) was obtained from National Institute of Standards and Technology Chemistry Web-Book (<http://webbook.nist.gov/chemistry>). Moreover, the correction energy of oxygen molecule ($E_{O_2(g)}^{correction}$) was added to reconcile the E_{vf-O} differences between the results achieved *via* computational method (generalized gradient approximation (GGA) functional) and real experimental results^{R4}.

With the re-calculated G_{vf-O} values from the surface AO layer, we can firstly screen the structurally unstable materials at 750 °C under reducing condition. Four materials (PBN, PSN, PBC, and PSC) demonstrated spontaneous oxygen vacancy formation ($G_{vf-O} < 0$) at the surface AO-terminated layers at 750 °C and $p(O_2) : 10^{-9}$ atm (reducing condition), implying phase decomposition at this specified condition. However, the other four materials (PBM, PSM, PBF, and PSF) displayed unspontaneous oxygen vacancy formation ($G_{vf-O} > 0$) at the surface AO-terminated layers at 750 °C and $p(O_2) : 10^{-9}$ atm (reducing condition) (Revised **Fig. 1a**).

(2) Phase reconstruction at 750 °C, $p(O_2) : 10^{-9}$ atm, $p(H_2) = 0.1$ atm, and $p(H_2O) = 0.01$ atm (BO₂ layer).

The phase reconstruction to Ruddlesden-Popper (R-P) perovskite or layered perovskite occurs by the reduction of BO₂ layers in the reduction atmosphere. Because the formation of oxygen vacancies at the BO₂ layer would include the hydrogen oxidation reaction (HOR, $O_{lattice} + H_2(g) \leftrightarrow H_2O(g)$) at the surface and the oxygen vacancy diffusion toward the BO₂ layer, we also calculated the G_{vf-O} from the BO₂ layer at the above specified condition (reducing condition) by using the following equations:

Equation R3, Equation R4, and Equation R5:

$$G_{vf-O} = E_{perov-defect} - E_{perov} + (\mu_{H_2O} - \mu_{H_2}) + E_a^{O_{vac} diffusion} \quad (R3)$$

$$\mu_{H_2O} = \left(\Delta H_{H_2O}^{exp} + E_{H_2(g)}^{DFT} + E_{H_2(g)}^{ZPE} + \frac{1}{2} (E_{O_2(g)}^{DFT} + E_{O_2(g)}^{ZPE} + E_{O_2(g)}^{correction}) \right) - TS_{H_2O(g)} + k_B T \ln \left(\frac{P_{H_2O}}{P_0} \right) \quad (R4)$$

$$\mu_{H_2} = E_{H_2(g)}^{DFT} + E_{H_2(g)}^{ZPE} - TS_{H_2(g)} + k_B T \ln \left(\frac{P_{H_2}}{P_0} \right) \quad (R5)$$

The zero-point energies were extracted from the previously calculated values^{R3}. The standard entropies of gas phase molecules, as well as the formation enthalpy of water ($(\Delta H_{H_2O}^{exp} : -286.41 \text{ kJ/mol})$, were obtained from National Institute of Standards and Technology Chemistry Web-Book (<http://webbook.nist.gov/chemistry>). The activation energy of oxygen vacancy diffusion ($E_a^{O_{vac} diffusion} : 0.95$ eV) was calculated from the electrochemical measurements (Arrhenius plot of area-specific resistance) of Pr_{0.4}Sr_{0.6}Fe_{0.875}Mo_{0.125}O_{3- δ} (PSFM) under H₂ condition^{R5}. Moreover, it was reported that the PBM perovskite undergoes phase transition to layered perovskite under specified reducing condition (At 750 °C, $p(O_2) : 10^{-9}$ atm, $p(H_2) = 0.1$ atm, and $p(H_2O) = 0.01$ atm). Thus, the G_{vf-O} for eight samples from the BO₂ layer was re-calculated at the above specified condition to find the appropriate samples that could demonstrate phase reconstruction to R-P perovskite.

In summary, we additionally calculated the G_{vf-O} of eight samples (in **Table R1**) from the surface AO and BO₂ layers to include the main factors for the oxygen vacancy formation (in this case, temperature and oxygen partial pressure). In the presented calculations, the borderlines for two criteria (phase decomposition and phase reconstruction) were both set to the G_{vf-O} of 0 eV for checking the spontaneity for two criteria

(Revised **Fig. 1a**). Furthermore, by calculation of spontaneous phase decomposition and/or reconstruction temperature where the $G_{v-f,O}$ value from surface AO layer and BO_2 layer becomes zero (**Fig. R1**), we can approximately determine whether the sample is stable at a specified temperature under reducing condition.

[References for the revision]

R1: Choi, S. *et al.* Highly efficient and robust cathode materials for low-temperature solid oxide fuel cells: $\text{PrBa}_{0.5}\text{Sr}_{0.5}\text{Co}_{2-x}\text{Fe}_x\text{O}_{5+\delta}$. *Sci. Rep.* **3**, 1–6 (2013).

R2: Lee, D. *et al.* Oxygen surface exchange kinetics and stability of $(\text{La,Sr})_2\text{CoO}_{4\pm\delta}/\text{La}_{1-x}\text{Sr}_x\text{MO}_{3-\delta}$ (M= Co and Fe) hetero-interfaces at intermediate temperatures. *J. Mater. Chem. A.* **3**, 2144–2157 (2015).

R3: Nørskov, J. K. *et al.* Origin of the overpotential for oxygen reduction at a fuel-cell cathode. *J. Phys. Chem. B.* **108**, 17886–17892 (2004).

R4: Wang, L., Maxisch, T., & Ceder, G. Oxidation energies of transition metal oxides within the GGA + U framework. *Phys. Rev. B.* **73**, 195107 (2006).

R5: Zhang, D. *et al.* Preparation and characterization of a redox-stable $\text{Pr}_{0.4}\text{Sr}_{0.6}\text{Fe}_{0.875}\text{Mo}_{0.125}\text{O}_{3-\delta}$ material as a novel symmetrical electrode for solid oxide cell application. *Int. J. Hydrog. Energy.* **45**, 21825–21835 (2020).

R6: Sengodan, S. *et al.* Layered oxygen-deficient double perovskite as an efficient and stable anode for direct hydrocarbon solid oxide fuel cells. *Nat. Mater.* **14**, 205–209 (2015).

[Table R1]

Table R1. Chemical composition and abbreviation of specimens used for the density functional theory (DFT) calculations.

Composition	Abbreviation
$\text{Pr}_{0.5}\text{Ba}_{0.5}\text{MnO}_{3-\delta}$	PBM
$\text{Pr}_{0.5}\text{Sr}_{0.5}\text{MnO}_{3-\delta}$	PSM
$\text{Pr}_{0.5}\text{Ba}_{0.5}\text{FeO}_{3-\delta}$	PBF (A-PBSF00 in the manuscript)
$\text{Pr}_{0.5}\text{Sr}_{0.5}\text{FeO}_{3-\delta}$	PSF (A-PBSF50 in the manuscript)
$\text{Pr}_{0.5}\text{Ba}_{0.5}\text{CoO}_{3-\delta}$	PBC
$\text{Pr}_{0.5}\text{Sr}_{0.5}\text{CoO}_{3-\delta}$	PSC
$\text{Pr}_{0.5}\text{Ba}_{0.5}\text{NiO}_{3-\delta}$	PBN
$\text{Pr}_{0.5}\text{Sr}_{0.5}\text{NiO}_{3-\delta}$	PSN

[Figure R1]

Figure R1. (a) Spontaneous phase decomposition temperature where the Gibbs free energy for oxygen vacancy formation (G_{vf-O}) at the surface AO layer becomes zero under $p(O_2)$: 10^{-9} atm. (b) Spontaneous reduction temperature where the G_{vf-O} at the BO_2 layer becomes zero under $p(O_2)$: 10^{-9} atm, P_{H_2} : 0.1 atm, and P_{H_2O} : 0.01 atm.

Added text in the revised manuscript:

Methods: Computational methods

Furthermore, the Gibbs free energy for oxygen vacancy formation of eight samples were calculated to include the temperature and oxygen partial pressure factors in the E_{vf-O} calculations. The equations used for the G_{vf-O} calculations from the surface AO and BO_2 layers in $Pr_{0.5}(Ba/Sr)_{0.5}TO_{3-\delta}$ ($T = Mn, Fe, Co,$ and Ni) are as follows:

$$G_{vf-O}(\text{AO layer}) = E_{perov-defect} - E_{perov} + \frac{1}{2}\mu_{O_2}$$

$$\mu_{O_2} = E_{O_2(g)}^{DFT} + E_{O_2(g)}^{ZPE} + E_{O_2(g)}^{correction} - TS_{O_2(g)} + k_B T \ln\left(\frac{P_{O_2}}{P_0}\right)$$

$$G_{vf-O}(\text{BO}_2 \text{ layer}) = E_{perov-defect} - E_{perov} + (\mu_{H_2O} - \mu_{H_2}) + E_a^{O_{vac} \text{ diffusion}}$$

$$\mu_{H_2O} = \left(\Delta H_{H_2O}^{exp} + E_{H_2(g)}^{DFT} + E_{H_2(g)}^{ZPE} + \frac{1}{2}(E_{O_2(g)}^{DFT} + E_{O_2(g)}^{ZPE} + E_{O_2(g)}^{correction}) \right) - TS_{H_2O(g)} + k_B T \ln\left(\frac{P_{H_2O}}{P_0}\right)$$

$$\mu_{H_2} = E_{H_2(g)}^{DFT} + E_{H_2(g)}^{ZPE} - TS_{H_2(g)} + k_B T \ln\left(\frac{P_{H_2}}{P_0}\right)$$

The $E_{perov-defect}$ and E_{perov} are the total energies of PrO -terminated (001) perovskite slab model with and without the oxygen vacancy, respectively. The μ_{O_2} , μ_{H_2} , and μ_{H_2O} are the Gibbs free energy of diatomic oxygen molecule, hydrogen, and water molecule, respectively. The $E_{O_2(g)}^{DFT}$ and $E_{H_2(g)}^{DFT}$ is the gas phase energy of ground state triplet O_2 molecule and hydrogen molecule, respectively. The zero point energies of oxygen molecule ($E_{O_2(g)}^{ZPE}$) and hydrogen molecule ($E_{H_2(g)}^{ZPE}$) were extracted from the previous calculated value⁵⁹. The standard entropy of gas phase oxygen ($S_{O_2(g)}$) was obtained from National Institute of Standards and Technology Chemistry Web-Book (<http://webbook.nist.gov/chemistry>). Moreover, the correction energy of oxygen molecule ($E_{O_2(g)}^{correction}$) was added to reconcile the E_{vf-O} differences between the

results achieved *via* computational method (generalized gradient approximation (GGA) functional) and real experimental results⁶⁰. The $p(\text{O}_2)$, $p(\text{H}_2)$, and $p(\text{H}_2\text{O})$ values are 10^{-9} , 0.1, and 0.01 atm, respectively. Under this specified condition (reducing condition), we assumed that the reduction of BO_2 layer occurred *via* two elementary steps: surface hydrogen oxidation reaction ($\text{O}_{\text{lattice}} + \text{H}_2(\text{g}) \leftrightarrow \text{H}_2\text{O}(\text{g})$) and oxygen vacancy diffusion toward the BO_2 layer. The activation energy of oxygen vacancy diffusion ($E_a^{\text{O}_{\text{vac}} \text{ diffusion}}$: 0.95 eV) was calculated from the electrochemical measurements (Arrhenius plot of area specific resistance) of $\text{Pr}_{0.4}\text{Sr}_{0.6}\text{Fe}_{0.875}\text{Mo}_{0.125}\text{O}_{3-\delta}$ (PSFM) material under H_2 condition⁶¹.

Added references in the revised manuscript:

59: Nørskov, J. K. *et al.* Origin of the overpotential for oxygen reduction at a fuel-cell cathode. *J. Phys. Chem. B.* **108**, 17886–17892 (2004).

60: Wang, L., Maxisch, T., & Ceder, G. Oxidation energies of transition metal oxides within the GGA + U framework. *Phys. Rev. B.* **73**, 195107 (2006).

61: Zhang, D. *et al.* Preparation and characterization of a redox-stable $\text{Pr}_{0.4}\text{Sr}_{0.6}\text{Fe}_{0.875}\text{Mo}_{0.125}\text{O}_{3-\delta}$ material as a novel symmetrical electrode for solid oxide cell application. *Int. J. Hydrog. Energy.* **45**, 21825–21835 (2020).

Added figure in the revised manuscript:

[Supplementary Figure 1]

Supplementary Figure 1. Calculated oxygen vacancy formation energies of $\text{Pr}_{0.5}(\text{Ba}/\text{Sr})_{0.5}\text{TO}_{3-\delta}$ ($T = \text{Mn}, \text{Fe}, \text{Co},$ and Ni) from AO (green bar) and BO_2 (purple bar) networks and the predicted phase change under reducing condition. Note that the Gibbs free energy for the oxygen vacancy formation ($G_{\text{vf-O}}$) was additionally calculated to contain the temperature and oxygen partial pressure factors in the $E_{\text{vf-O}}$ calculations.

Revised figure in the revised manuscript:

[Original figure]

Figure 1. (a) Calculated oxygen vacancy formation energies of $\text{Pr}_{0.5}(\text{Ba}/\text{Sr})_{0.5}\text{TO}_{3-\delta}$ ($T = \text{Mn}, \text{Fe}, \text{Co},$ and Ni) from AO (green bar) and BO_2 (purple bar) networks and the predicted phase change under reducing condition. (b) Schematic of the most stable structure configurations of $\text{Pr}_{0.5}(\text{Ba}/\text{Sr})_{0.5}\text{TO}_{3-\delta}$ ($T = \text{Mn}, \text{Fe}, \text{Co},$ and Ni) slab models used for the calculations of oxygen vacancy formation energy values from AO and BO_2 networks.

[Revised figure]

Figure 1. (a) Calculated Gibbs free energy for oxygen vacancy formation (G_{vf-O}) of $\text{Pr}_{0.5}(\text{Ba/Sr})_{0.5}\text{TO}_{3-\delta}$ (T = Mn, Fe, Co, and Ni) from the surface AO (green bar) and BO_2 (purple bar) networks and the predicted phase change under reducing condition. (b) Schematic of the most stable structure configurations of $\text{Pr}_{0.5}(\text{Ba/Sr})_{0.5}\text{TO}_{3-\delta}$ (T = Mn, Fe, Co, and Ni) slab models used for the calculations of G_{vf-O} values from the surface AO and BO_2 networks.

Revised text in the revised manuscript:

Abstract

[Original text]

Herein, the oxygen vacancy formation energies (E_{vf-O}) from PrO and TO_2 in $\text{Pr}_{0.5}(\text{Ba/Sr})_{0.5}\text{TO}_{3-\delta}$ (T = Mn, Fe, Co, and Ni) are proposed as the important factor in ...

[Revised text]

Herein, the Gibbs free energy for oxygen vacancy formation (G_{vf-O}) from PrO and TO_2 in $\text{Pr}_{0.5}(\text{Ba/Sr})_{0.5}\text{TO}_{3-\delta}$ (T = Mn, Fe, Co, and Ni) are proposed as the important factor in ...

Introduction

[Original text]

Here, we systematically calculated the oxygen vacancy formation energies (E_{vf-O}) of perovskite oxides with various cations to investigate the unprecedented factor affecting the phase reconstruction. The type of phase

reconstruction could be predicted with the E_{vf-O} value from PrO and TO_2 networks in $Pr_{0.5}(Ba/Sr)_{0.5}TO_{3-\delta}$ ($T = Mn, Fe, Co, \text{ and } Ni$), in which the most appropriate cations for the complete reconstruction to R-P perovskite were determined.

[Revised text]

Here, we systematically calculated the **Gibbs free energy for oxygen vacancy formation (G_{vf-O})** of perovskite oxides with various cations to investigate the unprecedented factor affecting the phase reconstruction. The type of phase reconstruction could be predicted with the G_{vf-O} value from PrO and TO_2 networks in $Pr_{0.5}(Ba/Sr)_{0.5}TO_{3-\delta}$ ($T = Mn, Fe, Co, \text{ and } Ni$), in which the most appropriate cations for the complete reconstruction to R-P perovskite were determined.

Results: Density functional theory calculations

[Original text-(1)]

To determine the unexplored factor for the phase reconstruction for the first time, the oxygen vacancy formation energies (E_{vf-O}) from AO (A-site) and BO_2 (B-site) networks were calculated for $Pr_{0.5}Ba_{0.5}TO_{3-\delta}$ and $Pr_{0.5}Sr_{0.5}TO_{3-\delta}$ ($T = Mn, Fe, Co, \text{ and } Ni$) perovskite oxides (**Figure 1**)³¹⁻³⁴.

[Revised text-(1)]

To determine the unexplored factor for the phase reconstruction for the first time, **the Gibbs free energy for oxygen vacancy formation (G_{vf-O}) and the oxygen vacancy formation energies (E_{vf-O}) from the surface AO (A-site) and BO_2 (B-site) networks were calculated for $Pr_{0.5}Ba_{0.5}TO_{3-\delta}$ and $Pr_{0.5}Sr_{0.5}TO_{3-\delta}$ ($T = Mn, Fe, Co, \text{ and } Ni$) perovskite oxides (**Fig. 1 and Supplementary Fig. 1**)³¹⁻³⁴.**

[Original text-(2)]

For the perovskite oxides to undergo phase reconstruction without phase decomposition under reducing condition, the A-site E_{vf-O} value should be higher than 1.5 eV. Moreover, the B-site E_{vf-O} value would be an important factor for determining the type of phase reconstruction. For instance, the B-site E_{vf-O} should be in the range of about 1.6 to 2.8 eV to demonstrate phase reconstruction to R-P perovskite in the reduction environment.

[Revised text-(2)]

For the perovskite oxides to undergo phase reconstruction without phase decomposition under reducing condition, the A-site G_{vf-O} value should be **positive (A-site $G_{vf-O} > 0$ eV)**. Moreover, the B-site G_{vf-O} value would be an important factor for determining the type of phase reconstruction. For instance, the B-site G_{vf-O} should be in the range of about **- 1.2 to 0 eV (- 1.2 eV < B-site $G_{vf-O} < 0$ eV)** to demonstrate phase reconstruction to R-P perovskite in the reduction environment.

Conclusion

[Original text]

In summary, this study successfully calculated $E_{\text{v-f-O}}$ value from PrO and TO₂ in Pr_{0.5}(Ba/Sr)_{0.5}TO_{3-δ} (T = Mn, Fe, Co, and Ni) as the key factor for identifying the type of the phase reconstruction.

[Revised text]

In summary, this study successfully calculated $G_{\text{v-f-O}}$ value from PrO and TO₂ in Pr_{0.5}(Ba/Sr)_{0.5}TO_{3-δ} (T = Mn, Fe, Co, and Ni) as the key factor for identifying the type of the phase reconstruction.

Comment 2. In Fig. 2b, the authors provided a comprehensive phase diagram based on *in-situ* XRD results. Take $x = 0.5$ as an example. Based on Fig. 1a, it should be a R-P perovskite phase. However, as shown in Fig. 2b, it is a simple perovskite (region I) after the reduction below 770 °C, simple perovskite + R-P perovskite + Fe metal (Region III) in 780-830 °C, and R-P perovskite + Fe metal (region IV) above 840 °C. So I wonder if the points represent equilibrium and stable states (which should be a basic requirement for a phase diagram), or just transient states due to the short reduction time? If the Pr_{0.5}Sr_{0.5}FeO₃ is reduced at 830 °C for a much longer time, can it transform to R-P perovskite + Fe metal (region IV)? If it is reduced at 850 °C for a much longer time, will the R-P perovskite decompose further?

Response to Comment 2: We appreciate the reviewer for suggesting us very interesting point. As we provided in **Fig. 1** and “**Response to Comment 1**” part, we first predicted the possible perovskite oxide candidates that could demonstrate phase reconstruction to Ruddlesden-Popper (R-P) perovskite under particular temperature and atmosphere. Afterwards, we experimentally exemplified the phase reconstruction behavior of Pr_{0.5}Ba_{0.5-x}Sr_xFeO_{3-δ} material ($x = 0, 0.1, 0.2, 0.3, 0.4, \text{ and } 0.5$) under reducing atmosphere *via in-situ* temperature & environment-controlled X-ray diffraction (XRD) measurements and mapped out the phase diagram (**Fig. 2b**). As listed in the “**Methods: in-situ phase reconstruction tendency evaluation**” part, the reduction temperatures were ranged from 700 °C to 870 °C and “**two hours**” were delayed at each temperature interval. To further confirm whether the plotted points in the phase diagram represent equilibrium and stable states (a basic requirement for a phase diagram) but not just a transient state due to short reduction time, we additionally conducted *in-situ* X-ray diffraction (XRD) measurements for Pr_{0.5}Sr_{0.5}FeO_{3-δ} (A-PBSF50) at 830 °C in the reduction environment with respect to time. The A-PBSF50 did not transform to R-P perovskite + Fe metal (region IV) but remained as simple perovskite + R-P perovskite + Fe metal (region III) at 830 °C even after 4 hours of reduction (enough reduction time for displaying phase transition behavior)^{R7-R9} (**Fig. R2**). This result implies that two hours of delay for each temperature interval (10 °C in this work) is sufficient for confirming the phase reconstruction behavior and the phase diagram represent equilibrium states for each temperature interval. Furthermore, we additionally reduced the A-PBSF50 material at 850 °C for 2 hours, 4 hours, and 100 hours under humidified H₂ atmosphere (3% H₂O) to check whether the completely phase-transitioned R-P perovskite will decompose or not after a much longer reduction time. As shown in **Fig. R3**, the R-P perovskite is not decomposed and there was no secondary phases even after 100-hour H₂-reduction process, implying that the R-P perovskite is structurally stable under reducing atmosphere. As a result, the phase diagram in **Fig. 2b** meets the basic requirements for the phase diagram and the temperature interval is much more important than the reduction time unless it is not just a

short-time reduction (*e.g.*, 10 minutes).

[References for the revision]

R7: Lv, H. *et al.* *In Situ* Investigation of Reversible Exsolution/Dissolution of CoFe Alloy Nanoparticles in a Co□Doped $\text{Sr}_2\text{Fe}_{1.5}\text{Mo}_{0.5}\text{O}_{6-\delta}$ Cathode for CO_2 Electrolysis. *Adv. Mater.* **32**, 1906193 (2020).

R8: Kim, K. *et al.* Mechanistic insights into the phase transition and metal ex-solution phenomena of $\text{Pr}_{0.5}\text{Ba}_{0.5}\text{Mn}_{0.85}\text{Co}_{0.15}\text{O}_{3-\delta}$ from simple to layered perovskite under reducing conditions and enhanced catalytic activity. *Energy Environ. Sci.* **14**, 873–882. (2021).

R9: Kim, S. *et al.* Self-Transforming Configuration Based on Atmospheric-Adaptive Materials for Solid Oxide Cells. *Sci. Rep.* **8**, 1–7. (2018).

[Figure R2]

Figure R2. *In-situ* powder X-ray diffraction patterns of $\text{Pr}_{0.5}\text{Sr}_{0.5}\text{FeO}_{3-\delta}$ (A-PBSF50) sintered at 1200 °C for 4 hours in air atmosphere (black), reduced at 830 °C for 2 hours in humidified H₂ (3% H₂O) (blue), and reduced at 830 °C for 4 hours in humidified H₂ (red).

[Figure R3]

Figure R3. X-ray diffraction patterns of $\text{Pr}_{0.5}\text{Sr}_{0.5}\text{FeO}_{3-\delta}$ (A-PBSF50) sintered at 1200 °C for 4 hours in air atmosphere (black), reduced at 850 °C for 2 hours in humidified H₂ atmosphere (3% H₂O) (purple), reduced at 850 °C for 4 hours in humidified H₂ atmosphere (3% H₂O, blue), and reduced at 850 °C for 100 hours in humidified H₂ atmosphere (red).

Comment 3. In Fig. 2d-e, how to determine the concentration of oxygen vacancy in the material, which is remarkably influenced by the temperature and oxygen partial pressure, before DFT calculation?

Response to Comment 3: We think that there was a misunderstanding on **Fig. 2d** and **2e**. In **Fig. 2d** and **2e**, we did not determine the concentration of oxygen vacancies of four materials before the density functional theory (DFT) calculations. We think that there was a confusion on **Fig. 2e** (left y-axis on **Fig. 2e** is not the oxygen vacancy concentration term, but the oxygen vacancy formation energy term) since the oxygen vacancy formation energy (left y-axis) looks like the “ $3-\delta$ ” term in $ABO_{3-\delta}$. Therefore, we feel sorry for giving confusion on this part, yet we did not determine the concentration of oxygen vacancies of four materials, but the oxygen vacancy formation energies of four materials (red color in **Fig. 2e**) were calculated *via* density functional theory (DFT) calculations.

Comment 4. Please provide the cross-sectional SEM image of the symmetric cell for the better understanding of the structure of the electrode layer, LDC interlayer, and their interface.

Response to Comment 4: We thank the reviewer for suggesting a great point of this manuscript. We agree that providing the cross-sectional scanning electron microscope (SEM) image of the $Pr_{0.5}Ba_{0.2}Sr_{0.3}FeO_{3-\delta}$ (A-PBSF30) symmetrical cell (cell configuration: A-PBSF30| $La_{0.4}Ce_{0.6}O_{2-\delta}$ (LDC)| $La_{0.9}Sr_{0.1}Ga_{0.8}Mg_{0.2}O_{3-\delta}$ (LSGM)|LDC|A-PBSF30) to help the readers better understand the structure of the fabricated cell. Accordingly, we added a supplementary figure displaying the cross-sectional SEM image of the A-PBSF30 symmetrical cell (**Supplementary Fig. 20**).

Added figure in the revised manuscript:

[Supplementary Figure 20]

Supplementary Figure 20. (a – b) Scanning electron microscope (SEM) images. (a) Cross-sectional SEM image of $Pr_{0.5}Ba_{0.2}Sr_{0.3}FeO_{3-\delta}$ (A-PBSF30) symmetrical cell with cell configuration of A-PBSF30| $La_{0.4}Ce_{0.6}O_{2-\delta}$ (LDC)| $La_{0.9}Sr_{0.1}Ga_{0.8}Mg_{0.2}O_{3-\delta}$ (LSGM)|LDC|A-PBSF30 sintered at 950 °C for 4 hours in air atmosphere. (b) SEM image displaying the microstructure of the interface between electrode/buffer layer and the electrolyte.

Revised text in the revised manuscript:

Experimental: Electrochemical performance measurements

[Original text]

The cells with configuration of “Electrode|LDC|LSGM|LDC|Electrode” were mounted on alumina tubes with ceramic adhesives (Ceramabond 552, Aremco) for electrochemical performance tests.

[Revised text]

The cells with configuration of “Electrode|LDC|LSGM|LDC|Electrode” were mounted on alumina tubes with ceramic adhesives (Ceramabond 552, Aremco) for electrochemical performance tests (**cross-sectional SEM image given in Supplementary Fig. 20**).

Comment 5. Why is the P_{\max} of the symmetric cell with PBSF50 electrodes lower than that of the symmetric cell with PBSF30 electrodes? Is it due to the lower catalytic activity of PBSF50 for the electrochemical oxidation of H_2 , or for the electrochemical reduction of O_2 , or both? The authors only provided very limited I-V results of the symmetric cell in SOFC and SOEC modes, from which we can hardly separate the activities of the anode and the cathode from each other. For the better evaluation of the material, at least the EIS of the symmetric cells in H_2 and O_2 should be provided to reveal its exact activities toward H_2 oxidation and O_2 reduction. The performance of the electrode should be compared with other best anode and cathode materials, not only the electrodes for the symmetric cells.

Response to Comment 5: We appreciate the reviewer for the constructive comment. We agree with the reviewer’s comment that without the electrochemical impedance spectroscopy (EIS) data, it is hard to distinguish whether the maximum power density (P_{\max}) difference of the symmetrical full-cells is derived from the catalytic activity for the electrochemical reduction of O_2 , or electrochemical oxidation of H_2 , or both. In this regard, the electrochemical impedance spectroscopy (EIS) measurements were performed and the corresponding Nyquist plots were demonstrated for $Pr_{0.5}Ba_{0.2}Sr_{0.3}FeO_{3-\delta}$ (A-PBSF30) & $Pr_{0.5}Sr_{0.5}FeO_{3-\delta}$ (A-PBSF50) symmetric half-cells (cell configuration: A-PBSF30 or A-PBSF50| $La_{0.9}Sr_{0.1}Ga_{0.8}Mg_{0.2}O_{3-\delta}$ (LSGM)|A-PBSF30 or A-PBSF50) supplying air to both sides to confirm the electro-catalytic activity for electrochemical O_2 reduction. Unfortunately, it is difficult to check the electro-catalytic activity for H_2 oxidation *via* symmetric half-cell supplying H_2 to both sides. The reason is that for the symmetric half-cell supplying air to both sides, an electrochemical reaction occurs ($\frac{1}{2}O_2 + 2e^- \rightarrow O^{2-}$) and instantaneous polarization occurs, while the electrochemical reaction does not occur when H_2 is only blown on both sides because there is no O^{2-} source and instantaneous polarization and the corresponding polarization resistance measurement is impossible (**Fig. R4**)^{R9,R10}. Thus, we compared the EIS for the symmetrical full-cells supplying air to the solid oxide fuel cell (SOFC) air electrode part and H_2 (fuel) to the SOFC fuel electrode part. Consequently, the polarization resistance difference for symmetrical full-cell was much larger (about $0.12 \Omega \text{ cm}^2$) than the symmetric half-cell supplying air to both sides (about $0.008 \Omega \text{ cm}^2$) at 800°C (**Fig. R5**). Therefore, the P_{\max} difference between A-PBSF30 symmetrical full-cell and A-PBSF50 symmetrical full-cell is affected by the electro-catalytic activity of both O_2 reduction and H_2 oxidation, but mostly derived from the

SOFC fuel electrode part (electrochemical H₂ oxidation) due to the difference in exsolved particle size and surface distribution of Fe metal particles. Furthermore, even though we compared the electrochemical performance of symmetrical full-cells with other reported papers, we also compared the electrochemical performance with well-known best anode and cathode material (**Table R2**). Despite the same electrolyte and similar electrolyte thickness, the A-PBSF30 symmetrical full-cell demonstrated higher electrochemical performance than the best-known SOFC cathodes and anodes.

[References for the revision-(1)]

R9: Sengodan, S. *et al.* Self-decorated MnO nanoparticles on double perovskite solid oxide fuel cell anode by *in situ* exsolution. *ACS. Sustain. Chem. Eng.* **5**, 9207–9213 (2017).

R10: Bian, L. *et al.* Co-free La_{0.6}Sr_{0.4}Fe_{0.9}Nb_{0.1}O_{3-δ} symmetric electrode for hydrogen and carbon monoxide solid oxide fuel cell. *Inter. J. Hydro. Energy.* **44**, 32210–32218 (2019).

[Figure R4]

Figure R4. (a – b) Schematic illustration of symmetric half-cell configurations supplying (a) air (O₂) to both sides and (b) fuel (H₂) to both sides.

[Figure R5]

Figure R5. (a – b) Electrochemical impedance spectroscopy (EIS) analysis. Impedance spectra of (a) Pr_{0.5}Ba_{0.2}Sr_{0.3}FeO_{3-δ} (A-PBSF30) and Pr_{0.5}Sr_{0.5}FeO_{3-δ} (A-PBSF50) symmetric half-cells supplying air to both

sides under open circuit condition and (b) A-PBSF30 and A-PBSF50 symmetrical full-cells supplying air to the air electrode and H₂ to the fuel electrode under open circuit condition.

[Table R2]

Table R2. Additional comparison of the electrochemical performance of La_{1-x}Sr_xGa_{1-y}Mg_yO_{3-δ} (x = 0.1 or 0.2, y = 0.17 or 0.2, LSGM) electrolyte-supported solid oxide fuel cells (SOFCs) reported in the recent literatures with the best-known SOFC anodes and cathodes.

Reference	Cell configuration (Air electrode Electrolyte Fuel electrode)	Electrolyte thickness (μm)	Maximum power density at 800 °C (W cm ⁻²)
Present study	A-PBSF30 LDC LSGM LDC R-PBSF30	~ 250	1.23
R11	NdBa _{0.5} Sr _{0.5} Co _{1.5} Fe _{0.5} O _{5+δ} – GDC LSGM LDC PrBaMn _{1.7} Co _{0.1} Ni _{0.2} O _{5+δ}	~ 250	0.97
R12	La _{0.6} Sr _{0.4} Co _{0.2} Fe _{0.8} O _{3-δ} – GDC LSGM SrGdNi _{0.2} Mn _{0.8} O _{4+δ} – GDC	~ 280	1.01
R13	La _{0.6} Sr _{0.4} Co _{0.2} Fe _{0.8} O _{3-δ} – GDC LDC LSGM LDC Sr _{0.95} Ti _{0.3} Fe _{0.63} Ni _{0.07} O _{3-δ}	~ 300	0.93
R14	PrBa _{0.5} Sr _{0.5} Fe _{1.5} Co _{0.5} O _{5+δ} – GDC LSGM LDC PrBaMn ₂ O _{5+δ} & Co-Fe alloy	~ 250	0.90

LSGM: La_{1-x}Sr_xGa_{1-y}Mg_yO_{3-δ} (x = 0.1 or 0.2, y = 0.17 or 0.2)

LDC: La_{0.4}Ce_{0.6}O_{2-δ}

GDC: Gd_{0.1}Ce_{0.9}O_{1.95} or Gd_{0.4}Ce_{0.6}O_{2-δ}

[References for the revision-(2) (**Table R2**)]

R11. Kwon, O. *et al.* Self-assembled alloy nanoparticles in a layered double perovskite as a fuel oxidation catalyst for solid oxide fuel cells. *J. Mater. Chem. A*, **6**, 15947–15953 (2018).

R12. Kim, K. J. *et al.* A highly active and Redox-Stable SrGdNi_{0.2}Mn_{0.8}O_{4±δ} anode with *in situ* exsolution of nanocatalysts. *ACS Catal.* **9**, 1172–1182 (2019).

R13. Zhu, T., Troiani, H. E., Mogni, L. V., Han, M., & Barnett, S. A. Ni-substituted Sr(Ti,Fe)O₃ SOFC anodes: achieving high performance via metal alloy nanoparticle exsolution. *Joule*, **2**, 478–496 (2018).

R14. Jun, A., Kim, J., Shin, J., & Kim, G. Achieving high efficiency and eliminating degradation in solid oxide electrochemical cells using high oxygen capacity perovskite. *Angew. Chem. Int. Ed.* **55**, 12512–12515 (2016).

Comment 6. For the co-electrolysis of H₂O/CO₂, the quantitative analysis of the product is necessary. How much H₂ and CO are in the syngas?

Response to Comment 6: We thank the reviewer for the valuable comment. The quantitative analysis of the products *via* only H₂O or CO₂ electrolysis is possible by the current density value assuming 100% current efficiency^{R15,R16}. Nevertheless, in the case of co-electrolysis of H₂O and CO₂, gas chromatography (GC) method should be utilized for both qualitative and quantitative analysis of the synthetic gas (H₂ + CO) product. Thus, we performed the *in-operando* quantitative GC profiles of outlet gas during co-electrolysis of H₂O and CO₂ at 800 °C and 1.5 V (**Supplementary Fig. 19**)^{R17}. The amount of produced H₂ and CO during co-electrolysis at 800 °C for the A-PBSF30 symmetrical cell were measured to be 0.18 ml min⁻¹ (0.50 ml min⁻¹ cm⁻²) and 3.89 ml min⁻¹ (or 10.81 ml min⁻¹ cm⁻²), respectively. These results indicate that the A-PBSF30 symmetrical cell could efficiently produce synthetic gas (H₂ & CO) during co-electrolysis.

[References for the revision]

R15: Kim, J. *et al.* Hybrid-solid oxide electrolysis cell: A new strategy for efficient hydrogen production. *Nano Energy*. **44**, 121–126 (2018).

R16: Song, Y., Zhang, X., Xie, K., Wang, G., & Bao, X. High-temperature CO₂ electrolysis in solid oxide electrolysis cells: Developments, Challenges, and Prospects. *Adv. Mater.* **31**, 1902033 (2019).

R17: Kim, C. *et al.* Highly efficient CO₂ utilization via aqueous zinc–or aluminum–CO₂ systems for hydrogen gas evolution and electricity production. *Angew. Chem. Int. Ed.* **58**, 9506–9511 (2019).

Added figures in the revised manuscript:

[Supplementary Figure 19]

Supplementary Figure 19. (a – b) The *in-operando* qualitative gas chromatography (GC) profiles of (a) H₂ gas and (b) CO & CO₂ gases before (Reference, black) and during co-electrolysis at 800 °C and 1.5 V for the Pr_{0.5}Ba_{0.2}Sr_{0.3}FeO_{3- δ} (A-PBSF30) symmetrical cell (Co-electrolysis, red). The quantitative analysis of the generated gas products (H₂ & CO) were determined by the additional measurement of the actual flow rate of the reference gas and the outlet gas during the co-electrolysis. The exact volume of gases were calibrated through a bubble flow-meter.

Added text in the revised manuscript:

Results: Electrochemical performance

The *in-operando* quantitative analysis of the synthetic gas products (H₂ & CO) was further investigated *via* gas chromatography (GC) profiles for the A-PBSF30 symmetrical cell at 800 °C during co-electrolysis of H₂O and CO₂ (**Supplementary Fig. 19**). The amount of generated H₂ and CO were measured to be 0.50 ml min⁻¹ cm⁻² and 10.81 ml min⁻¹ cm⁻², respectively, implying that the A-PBSF30 symmetrical cell could efficiently produce synthetic gas during co-electrolysis.

Experimental: Electrochemical performance measurements

The *in-operando* quantitative analysis of the generated synthetic gas (H₂ & CO) during co-electrolysis of CO₂ and H₂O (Ratio of CO₂:H₂:H₂O = 45:45:10) for the A-PBSF30 symmetrical cell (@ 800 °C & 1.5 V) was demonstrated by the gas chromatograph (Agilent 7820A GC instrument) with a thermal conductivity detector (TCD) and a packed column (Agilent carboxen 1000).

Added reference in the revised manuscript:

50. Kim, C. *et al.* Highly efficient CO₂ utilization *via* aqueous zinc–or aluminum–CO₂ systems for hydrogen gas evolution and electricity production. *Angew. Chem. Int. Ed.* **58**, 9506–9511 (2019).

Revised text in the revised manuscript:

Results: Electrochemical performance

[Original text]

Together with superior electrochemical performance, ...

[Revised text]

Together with superior electrochemical performance **and efficient synthetic gas production**, ...

Answers to Reviewer #2

General Comments:

A comprehensive understanding of key parameters affecting the phase reconstruction from single perovskite to R-P perovskite is an interesting topic since most perovskite show phase reconstruction during reduction. But the main conclusion cannot be supported by the experimental results. This work is not suitable to be published in its present form in this journal. The questionable points are listed below.

Response to General Comments: We sincerely appreciate the reviewer for providing constructive comments to improve the quality of our manuscript. We endeavored to reflect the reviewer's comments as much as possible. The detailed point-by-point responses to the comments of the reviewer is provided below.

Comment 1. The author declares that oxygen vacancy formation energies (E_{v-f}) from PrO and TO_2 in $\text{Pr}_{0.5}(\text{Ba}/\text{Sr})_{0.5}\text{TO}_{3-\delta}$ ($T = \text{Mn, Fe, Co, and Ni}$) are proposed as the important factor in determining the type of phase reconstruction in perovskites. However, this conclusion is only supported by $T = \text{Fe}$, how about the Mn for instance? The experiments on sample of $T = \text{Mn}$ should be added.

Response to Comment 1: We appreciate the reviewer for the constructive comment. According to the reviewer's suggestion, we additionally synthesized $\text{Pr}_{0.5}\text{Sr}_{0.5}\text{MnO}_{3-\delta}$ material yet did not synthesize $\text{Pr}_{0.5}\text{Ba}_{0.5}\text{MnO}_{3-\delta}$ (PBM) material since the phase reconstruction behavior of the PBM material under reducing condition is already reported^{R6,R18,R19}. The PBM material displayed phase reconstruction from simple cubic & hexagonal perovskite phase ($\text{ABO}_{3-\delta}$) to layered perovskite ($\text{A}_2\text{B}_2\text{O}_{5+\delta}$) phase in the reduction environment (**Fig. R6**)^{R18}. However, the $\text{Pr}_{0.5}\text{Sr}_{0.5}\text{MnO}_{3-\delta}$ material demonstrated phase reconstruction from simple perovskite to K_2NiF_4 -type Ruddlesden-Popper perovskite ($\text{A}_2\text{BO}_{4+\delta}$) along with MnO exsolution under reducing condition (**Supplementary Fig. 2**). Therefore, not only from reference, we also experimentally demonstrated that the $\text{Pr}_{0.5}\text{Sr}_{0.5}\text{MnO}_{3-\delta}$ material undergo phase reconstruction from simple perovskite to R-P perovskite under reducing condition.

[References for the revision]

R6: Sengodan, S. *et al.* Layered oxygen-deficient double perovskite as an efficient and stable anode for direct hydrocarbon solid oxide fuel cells. *Nat. Mater.* **14**, 205–209 (2015).

R18: Kwon, O. *et al.* Exsolution trends and co-segregation aspects of self-grown catalyst nanoparticles in perovskites. *Nat. Commun.* **8**, 1–7 (2017).

R19: Sun, Y. F. *et al.* New opportunity for *in situ* exsolution of metallic nanoparticles on perovskite parent. *Nano Lett.* **16**, 5303–5309. (2016).

Added figure in the revised manuscript:

[Supplementary Figure 2]

Supplementary Figure 2. (a – b) X-ray diffraction patterns of (a) $\text{Pr}_{0.5}\text{Sr}_{0.5}\text{MnO}_{3-\delta}$ sintered at 1200 °C for 4 hours in air atmosphere (black) and reduced at 850 °C for 4 hours in humidified H₂ atmosphere (3% H₂O, red) and (b) $\text{Pr}_{0.5}\text{Sr}_{0.5}\text{FeO}_{3-\delta}$ (A-PBSF50) sintered at 1200 °C for 4 hours in air atmosphere (black) and reduced at 850 °C for 4 hours in humidified H₂ atmosphere (red).

[Figure R6]

Figure R6. (a – b) X-ray diffraction patterns of (a) $\text{Pr}_{0.5}\text{Ba}_{0.5}\text{MnO}_{3-\delta}$ (S-PBMO) sintered at 950 °C for 4 hours in air atmosphere and (b) $\text{PrBaMn}_2\text{O}_{5+\delta}$ (L-PBMO) reduced at 800 °C for 4 hours in humidified H₂ atmosphere (3% H₂O). The phase change from simple cubic & hexagonal perovskite (left) to layered perovskite (right) was observed after the reduction process^{R18}.

Revised text in the revised manuscript:

Results: Density functional theory calculations

[Original text]

Considering the aforementioned results, only $\text{Pr}_{0.5}\text{Sr}_{0.5}\text{MnO}_{3-\delta}$ (PSM) and $\text{Pr}_{0.5}\text{Sr}_{0.5}\text{FeO}_{3-\delta}$ (PSF) are the possible candidates for the phase reconstruction to R-P perovskite in this study.

[Revised text]

Considering the aforementioned results and the experimental data, only $\text{Pr}_{0.5}\text{Sr}_{0.5}\text{MnO}_{3-\delta}$ and $\text{Pr}_{0.5}\text{Sr}_{0.5}\text{FeO}_{3-\delta}$.

δ (A-PBSF50) are the possible candidates for the phase reconstruction to R-P perovskite in this study (**Supplementary Fig. 2**).

Method: Material synthesis

[Original text-(1)]

$\text{Pr}_{0.5}\text{Ba}_{0.5-x}\text{Sr}_x\text{FeO}_{3-\delta}$ samples ($x = 0, 0.1, 0.2, 0.3, 0.4$ and 0.5 , abbreviated as PBSF in **Supplementary Table 1**) were synthesized by the Pechini method. Stoichiometric amounts of $\text{Pr}(\text{NO}_3)_3 \cdot 6\text{H}_2\text{O}$ (Aldrich, 99.9%, metal basis), $\text{Ba}(\text{NO}_3)_2$ (Aldrich, 99+%), $\text{Sr}(\text{NO}_3)_2$ (Aldrich, 99+%), and $\text{Fe}(\text{NO}_3)_3 \cdot 9\text{H}_2\text{O}$ (Aldrich, 98+%) nitrate salts were dissolved in distilled water with the addition of quantitative amounts of citric acid and poly-ethylene glycol.

[Revised text-(1)]

$\text{Pr}_{0.5}\text{Ba}_{0.5-x}\text{Sr}_x\text{FeO}_{3-\delta}$ samples ($x = 0, 0.1, 0.2, 0.3, 0.4$ and 0.5 , abbreviated as PBSF in **Supplementary Table 1**) and $\text{Pr}_{0.5}\text{Sr}_{0.5}\text{MnO}_{3-\delta}$ were synthesized by the Pechini method. For PBSF materials, stoichiometric amounts of $\text{Pr}(\text{NO}_3)_3 \cdot 6\text{H}_2\text{O}$ (Aldrich, 99.9%, metal basis), $\text{Ba}(\text{NO}_3)_2$ (Aldrich, 99+%), $\text{Sr}(\text{NO}_3)_2$ (Aldrich, 99+%), and $\text{Fe}(\text{NO}_3)_3 \cdot 9\text{H}_2\text{O}$ (Aldrich, 98+%) nitrate salts were dissolved in distilled water with the addition of quantitative amounts of citric acid and poly-ethylene glycol, while for $\text{Pr}_{0.5}\text{Sr}_{0.5}\text{MnO}_{3-\delta}$ material, stoichiometric amounts of $\text{Pr}(\text{NO}_3)_3 \cdot 6\text{H}_2\text{O}$ (Aldrich, 99.9%, metal basis), $\text{Sr}(\text{NO}_3)_2$ (Aldrich, 99+%), and $\text{Mn}(\text{NO}_3)_2 \cdot 4\text{H}_2\text{O}$ (Aldrich, 97+%) nitrate salts were dissolved in distilled water with the addition of quantitative amounts of citric acid and poly-ethylene glycol.

Method: Structural characterization

[Original text]

The crystal structure of the $\text{Pr}_{0.5}\text{Ba}_{0.5-x}\text{Sr}_x\text{FeO}_{3-\delta}$ ($x = 0, 0.3$, and 0.5) samples after heat-treated in two different environmental conditions (sintered at 1200°C for 4 hours in air environment and reduced at 850°C for 4 hours in humidified H_2 environment (3% H_2O)) were first identified by powder X-ray diffraction (XRD) patterns (Rigaku diffractometer, Cu $K\alpha$ radiation, 40 kV, 40 mA) in the 2θ range of $20^\circ < 2\theta < 60^\circ$.

[Revised text]

The crystal structure of the $\text{Pr}_{0.5}\text{Ba}_{0.5-x}\text{Sr}_x\text{FeO}_{3-\delta}$ ($x = 0, 0.3$, and 0.5) and $\text{Pr}_{0.5}\text{Sr}_{0.5}\text{MnO}_{3-\delta}$ samples after heat-treated in two different environmental conditions (sintered at 1200°C for 4 hours in air environment and reduced at 850°C for 4 hours in humidified H_2 environment (3% H_2O)) were first identified by powder X-ray diffraction (XRD) patterns (Bruker diffractometer (LYNXEYE 1D detector), Cu $K\alpha$ radiation, 40 kV, 40 mA) in the 2θ range of $20^\circ < 2\theta < 60^\circ$.

Comment 2. The Sr doping can decrease the E_{recon} , but the difference between $x = 0$ and $x = 0.5$ are only 0.3 eV, it is hard to understand that such a small difference can impose a big influence to phase reconstruction. The author should also calculate the formation energy of RP phase with different Sr contents.

Response to Comment 2: We appreciate the reviewer for the helpful comment. The reviewer considered the phase reconstruction energy (E_{recon}) difference of 0.3 eV as somewhat low and was questionable that this small difference could induce a big influence in terms of phase reconstruction to R-P perovskite. This was a great chance to re-consider our mistake of energy unit in **Fig. 2d** since the actual unit was not eV but eV/unit cell. Hence, we are sorry for giving the reviewer a little confusion of the E_{recon} value during the energy normalization by the unit cell. To eliminate the confusion of this data, we changed the unit from eV/unit cell to eV. Thus, the real difference between $x = 0$ sample ($\text{Pr}_{0.5}\text{Ba}_{0.5}\text{FeO}_{3-\delta}$, A-PBSF00) and $x = 0.5$ sample ($\text{Pr}_{0.5}\text{Sr}_{0.5}\text{FeO}_{3-\delta}$, A-PBSF50) is 3.94 eV or 380.04 kJ/mol (Revised **Fig. 2d**). This obvious E_{recon} difference between A-PBSF00 and A-PBSF50 is meaningful since this much difference can impose a big influence on the phase reconstruction. Also, we agree with the reviewer's comment that it would be better to show more calculated E_{recon} values with more different Sr^{2+} contents. We additionally calculated the E_{recon} values at $x \approx 0.06$ and $x \approx 0.13$, and then verified that E_{recon} decreases as x increases (**Supplementary Fig. 7**).

Revised figure in the revised manuscript:

[Original figure]

Figure 2d. Calculated profiles of the relative total energy required for the phase reconstruction from simple perovskite to R-P perovskite as a function of Sr^{2+} concentration in four models.

[Revised figure]

Figure 2d. Calculated profiles of the relative total energy required for the phase reconstruction from simple perovskite to R-P perovskite as a function of Sr²⁺ concentration in four models.

Added figures in the revised manuscript:

[Supplementary Figure 7]

Supplementary Figure 7. (a – b) Calculated profiles of the relative total energy required for the phase reconstruction from simple perovskite to R-P perovskite as a function of Sr²⁺ concentration (a) in six models in terms of eV and (b) in four models in terms of eV/unit cell.

Revised text in the manuscript:

Results: Effect of Sr²⁺ concentration on computational calculations

[Original text]

The E_{recon} decreases with increasing Sr²⁺ concentration in PBSF, indicating that the incorporation of Sr²⁺ into Ba²⁺ site promotes the phase reconstruction to R-P perovskite.

[Revised text]

The E_{recon} decreases with increasing Sr²⁺ concentration in PBSF, indicating that the incorporation of Sr²⁺ into Ba²⁺ site promotes the phase reconstruction to R-P perovskite (**Supplementary Fig. 7**).

Methods: Computational methods

[Original text]

The relative energies required for the phase reconstruction from simple perovskite to n = 1 R-P perovskite (E_{recon}) of four model structures with different Sr²⁺ concentration were calculated using the total energy difference between simple perovskite and n = 1 R-P perovskite by the following equation:

[Revised text]

The relative energies required for the phase reconstruction from simple perovskite to n = 1 R-P perovskite (E_{recon}) of **six** model structures with different Sr²⁺ concentration were calculated using the total energy difference between simple perovskite and n = 1 R-P perovskite by the following equation:

Comment 3. The assignation of Fe in X-ray photoelectron spectra is questionable. The noted binding energy value of Fe is too high.

Response to Comment 3: We thank the reviewer for the prudent comment. We stated in the caption of **Supplementary Fig. 10** (before the revision) that the “binding energy peaks of 727.2, 725.6, 723.5, and 720.0 eV are corresponded to Fe⁴⁺, Fe³⁺, Fe²⁺, and Fe⁰ 2p_{1/2}, respectively. There are two main binding energy regions for Fe spectra: Fe 2p_{1/2} and Fe 2p_{3/2}. To support our assignation of Fe 2p_{1/2} in X-ray photoelectron spectra (XPS), we re-checked more references (For reference, the binding energy peaks of 710.8, 709.6, and 706.7 eV are corresponded to Fe³⁺, Fe²⁺, and Fe⁰ 2p_{3/2})^{R20-R23}. Moreover, since the reviewer #3 gave us a valuable comment that the identification of Fe⁴⁺ from only XPS fitting is daring, we also conducted X-ray absorption fine structure (XAFS) measurements (**Figs. 4a, 4b, and 4c**). Along with the identification of Fe⁴⁺ from the XAFS measurements, the increase in Fe-Fe shell intensity after reduction process precisely proves the exsolution of Fe⁰ metal.

[References for the revision]

R20: Kim, J. *et al.* Synergistic coupling derived cobalt oxide with nitrogenated holey two-dimensional matrix as an efficient bifunctional catalyst for metal–air batteries. *ACS nano*. **13**, 5502–5512 (2019).

R21: Mahmood, J. *et al.* Fe@C₂N: A highly-efficient indirect-contact oxygen reduction catalyst. *Nano Energy*. **44**, 304–310 (2018).

R22: Peng, D. L. *et al.* X-ray diffraction and X-ray photoelectron spectra of Fe-Cr-N films deposited by DC reactive sputtering. *J. Mater. Sci.* **34**, 4623–4628 (1999).

R23: Wang, Z. *et al.* Elevating the d-Band Center of Six-Coordinated Octahedrons in Co₉S₈ through Fe Incorporated Topochemical Deintercalation. *Adv. Energy Mater.* **11**, 2003023 (2021).

Added figure in the revised manuscript:

[Figure 4]

Figure 4. (a – e) Oxidation state characterization. (a – b) Fe K-edge X-ray absorption near-edge structure (XANES) spectra of $\text{Pr}_{0.5}\text{Ba}_{0.2}\text{Sr}_{0.3}\text{FeO}_{3-\delta}$ (A-PBSF30), $(\text{Pr}_{0.5}\text{Ba}_{0.2}\text{Sr}_{0.3})_2\text{FeO}_{4+\delta}$ (R-PBSF30) with two references (Fe foil and Fe_2O_3). (c) Fourier-transformed Fe K-edge extended X-ray absorption fine structure (EXAFS) spectra of A-PBSF30 and R-PBSF30. (d – e) X-ray photoelectron spectra (XPS) of Fe 2p_{1/2} for (d) A-PBSF30 and (e) R-PBSF30.

Added text in the revised manuscript:

Methods: Structural characterization

The X-ray absorption fine structure (XAFS) spectra of Fe K-edge for A-PBSF30, R-PBSF30, and two references (Fe foil and Fe_2O_3 powder) were measured on ionization detectors under fluorescence mode at the Pohang Accelerator Laboratory (PAL, 6D extended XAFS (EXAFS)). The XAFS and Fourier-transformed (FT) EXAFS spectra analysis were performed using the Athena (Demeter) program.

Comment 4. XRD refinement should give the phase ratios of RP to Fe metal to confirm the equation (2).

Response to Comment 4: We thank the reviewer for the insightful comment. The equation (2) that we provided in the manuscript was based on the stoichiometric chemistry nature of the phase reconstruction from $\text{Pr}_{0.5}\text{Ba}_{0.2}\text{Sr}_{0.3}\text{FeO}_{3-\delta}$ (A-PBSF30) to $(\text{Pr}_{0.5}\text{Ba}_{0.2}\text{Sr}_{0.3})_2\text{FeO}_{4+\delta}$ – Fe metal (R-PBSF30) in the reduction environment. The reviewer stated that the phase ratio between Ruddlesden-Popper (R-P) perovskite and exsolved Fe metal should be given from the X-ray diffraction (XRD) refinement profile to confirm equation (2). In general, the phase ratio of mixed phases is determined the fractional scale factor^{R24,R25}, in which the phase ratio is determined through the intensity ratio between two known phases (For reference, the sum of phase fraction for mixed phases is 100%). However, in our position, the intensity ratio between the R-P phase and the Fe metal phase after the exsolution phenomenon is hard to determine because it is hard to determine the fractional scale factor assuming 100% Fe metal exsolution. In addition, the amount of exsolved particles is hard to acquire from the intensity of XRD patterns, thus it is difficult to obtain the phase fraction of exsolved metal particles just from XRD spectra. As a result, since the phase ratio of R-P phase and exsolved Fe metal could not be determined from the XRD refinement data, we could only suggest equation (2) only based on the stoichiometric chemistry.

[References for the revision]

R24: Lim, C. *et al.* Influence of Ca-doping in layered perovskite $\text{PrBaCo}_2\text{O}_{5+\delta}$ on the phase transition and cathodic performance of a solid oxide fuel cell. *J. Mater. Chem. A*, **4**, 6479–6486 (2016).

R25: Song, Y. *et al.* Self-assembled triple-conducting nanocomposite as a superior protonic ceramic fuel cell cathode. *Joule*, **3**, 2842–2853 (2019).

[Equation (2) in the manuscript]

Comment 5. In **Supplementary Figure 2b**, the XRD peak at 33° of R-PBSF30 looks much higher than the standard peak, why? And it is also not well refined in the XRD refinement (**Supplementary Figure 3**).

Response to Comment 5: We thank for the reviewer's prominent comment for improving the quality of this manuscript. Because the intensity ratio for **Supplementary Fig. 2b** and **Supplementary Fig. 3** (used for X-ray diffraction (XRD) refinement) is different, we re-synthesized the same $\text{Pr}_{0.5}\text{Ba}_{0.2}\text{Sr}_{0.3}\text{FeO}_{3-\delta}$ (A-PBSF30) sample *via* the Pechini method and then reduced the sample under humidified H_2 atmosphere (3% H_2O) to check whether this phenomenon was originated from experimental error or preferred orientation effect (or preferential growth of certain crystal planes during the synthesis procedure)^{R26}. As shown in the XRD measurement in **Supplementary Fig. 4b** (was **Supplementary Fig. 2b**), the slight intensity difference between two main peaks for the Ruddlesden-Popper perovskite phase (2 theta $\approx 31.5^\circ$ and 33°) was displayed between $(\text{Pr}_{0.5}\text{Ba}_{0.2}\text{Sr}_{0.3})_2\text{FeO}_{4+\delta}$ – Fe metal (R-PBSF30) and $(\text{Pr}_{0.5}\text{Sr}_{0.5})_2\text{FeO}_{4+\delta}$ – Fe metal (R-PBSF50) even after re-synthesis of samples. Accordingly, there is a slight preferential growth of certain crystal planes during the reduction procedure between R-PBSF30 and R-PBSF50 in the (110) plane (about 2 theta $\approx 33^\circ$). Furthermore, we also re-performed the XRD refinement for the re-synthesized R-PBSF30 using the GSAS II program along with using the crystallographic information file (CIF). The small difference in **Supplementary Fig. 5** (was **Supplementary Fig. 3**) indicates that the R-PBSF30 is well-refined^{R27,R28}. This valuable comment gave us a great opportunity to re-consider our Supplementary Figures in terms of XRD measurements and XRD refinement.

[References for the revision]

R26: Lv, Y., Lin, J., Peng, S., Zhang, L., & Yu, L. Effective ways to enhance the photocatalytic activity of ZnO nanopowders: high crystalline degree, more oxygen vacancies, and preferential growth. *New. J. Chem.* **43**, 19223–19231 (2019).

R27: Kim, J. *et al.* Chemically Stable Perovskites as Cathode Materials for Solid Oxide Fuel Cells: La□ Doped $\text{Ba}_{0.5}\text{Sr}_{0.5}\text{Co}_{0.8}\text{Fe}_{0.2}\text{O}_{3-\delta}$. *ChemSusChem.* **7**, 1669–1675 (2014).

R28: Choi, S., Park, S., Shin, J., & Kim, G. The effect of calcium doping on the improvement of performance and durability in a layered perovskite cathode for intermediate-temperature solid oxide fuel cells. *J. Mater. Chem. A.* **3**, 6088–6095 (2015).

Revised Figure in the manuscript:

[Original Figure-(1)]

Supplementary Figure 2. (a – b) X-ray diffraction patterns of $\text{Pr}_{0.5}\text{Ba}_{0.5-x}\text{Sr}_x\text{FeO}_{3-\delta}$ ($x = 0, 0.3, \text{ and } 0.5$) (a) sintered at 1200°C for 4 hours in air atmosphere and (b) reduced at 850°C for 4 hours in H_2 atmosphere.

[Revised Figure-(1)]

Supplementary Figure 4. (a – b) X-ray diffraction patterns of $\text{Pr}_{0.5}\text{Ba}_{0.5-x}\text{Sr}_x\text{FeO}_{3-\delta}$ ($x = 0, 0.3, \text{ and } 0.5$) (a) sintered at 1200°C for 4 hours in air atmosphere and (b) reduced at 850°C for 4 hours in H_2 atmosphere.

[Original Figure-(2)]

Supplementary Figure 3. X-ray diffraction (XRD) Rietveld refinement data of $(\text{Pr}_{0.5}\text{Ba}_{0.2}\text{Sr}_{0.3})_2\text{FeO}_{4+\delta}$ – Fe metal (R-PBSF30) reduced at 850°C for 4 hours in humidified H_2 atmosphere (3% H_2O). The R-PBSF30

exhibited a tetragonal structure, space group $I4/mmm$ with lattice parameters of $a = b = 3.881$ and $c = 12.713$ Å, which is clearly different to the simple perovskite $\text{Pr}_{0.5}\text{Ba}_{0.2}\text{Sr}_{0.3}\text{FeO}_{3-\delta}$ (A-PBSF30) with space group $Pm-3m$ (Supplementary Figure 1d).

[Revised Figure-(2)]

Supplementary Figure 5. X-ray diffraction (XRD) Rietveld refinement data of $(\text{Pr}_{0.5}\text{Ba}_{0.2}\text{Sr}_{0.3})_2\text{FeO}_{4+\delta}$ – Fe metal (R-PBSF30) reduced at 850 °C for 4 hours in humidified H_2 atmosphere (3% H_2O). The R-PBSF30 exhibited a tetragonal structure, space group $I4/mmm$ with lattice parameters of $a = b = 3.879$ and $c = 12.704$ Å, which is clearly different to the simple perovskite $\text{Pr}_{0.5}\text{Ba}_{0.2}\text{Sr}_{0.3}\text{FeO}_{3-\delta}$ (A-PBSF30) with space group $Pm-3m$ (Supplementary Fig. 3d).

Comment 6. The conductivity of the RP phase is much low and the highest value is only 2 S cm^{-1} . Why could such high cell performance be obtained?

Response to Comment 6: We thank for the reviewer's valuable comment. Unlike the least electrical conductivity value requirement in air atmosphere (at least 100 S cm^{-1})^{R29,R30} the least electrical conductivity value requirement for samples under reducing atmosphere is 1 S cm^{-1} ^{R31-R33}. Hence, the electrical conductivity value of the Ruddlesden-Popper (R-P) perovskites in this work under reducing condition is sufficient to be utilized as solid oxide fuel cell fuel electrodes. Moreover, great electrochemical performance (or high cell performance) for solid oxide fuel cells (SOFCs) is not only affected by electrical conductivity, but also the catalytic activity of the electrode material. For instance, even though two materials, $\text{Pr}_{0.5}\text{Ba}_{0.5}\text{MnO}_{3-\delta}$ (PBM) and $\text{Pr}_{0.5}\text{Ba}_{0.5}\text{Mn}_{0.85}\text{T}_{0.15}\text{O}_{3-\delta}$ (PBMT, T = Co and/or Ni) would display similar electrical conductivity value under reducing atmosphere (but there is no electrical conductivity value comparison between PBM and PBMT), Co and/or Ni doping into PBM will not significantly increase the electrical conductivity value, according to the Zener double exchange mechanism^{R34,R35}. Despite the similarly expected electrical conductivity value under reducing condition, the maximum power density for single cell with PBMT fuel electrode at 800 °C is about two-fold compared to single cell with PBM fuel electrode^{R6,R18}. Therefore, if the electrical conductivity value is shown to a certain extent, the development of SOFC fuel electrode material with excellent catalytic activity toward fuel oxidation reaction is essential for demonstrating great electrochemical performance. In this work, excellent electrochemical performance for the $\text{Pr}_{0.5}\text{Ba}_{0.2}\text{Sr}_{0.3}\text{FeO}_{3-\delta}$ (A-PBSF30) symmetrical cell was attributed to the exsolution of abundant Fe metal particles possessing remarkable electro-catalytic activity toward H_2 oxidation.

[References for the revision]

R6: Sengodan, S. *et al.* Layered oxygen-deficient double perovskite as an efficient and stable anode for direct hydrocarbon solid oxide fuel cells. *Nat. Mater.* **14**, 205–209 (2015).

R18: Kwon, O. *et al.* Exsolution trends and co-segregation aspects of self-grown catalyst nanoparticles in perovskites. *Nat. Commun.* **8**, 1–7 (2017).

R29: Kim, H., Joo, S., Kwon, O., Choi, S., & Kim, G. Cobalt-free $\text{Pr}_{0.5}\text{Ba}_{0.4}\text{Sr}_{0.1}\text{FeO}_{3-\delta}$ as a highly efficient cathode for commercial YSZ-supported solid oxide fuel cell. *ChemElectroChem.* **7**, 4378–4382 (2020).

R30: Jun, A. *et al.* Correlation between fast oxygen kinetics and enhanced performance in Fe doped layered perovskite cathodes for solid oxide fuel cells. *J. Mater. Chem. A.* **3**, 15082–15090 (2015).

R31: Bastidas, D. M., Tao, S., & Irvine, J. T. A symmetrical solid oxide fuel cell demonstrating redox stable perovskite electrodes. *J. Mater. Chem.* **16**, 1603–1605 (2006).

R32: Tao, S., & Irvine, J. T. A redox-stable efficient anode for solid-oxide fuel cells. *Nat. Mater.* **2**, 320–323 (2003).

R33: Huang, Y. H., Dass, R. I., Xing, Z. L., & Goodenough, J. B. Double perovskites as anode materials for solid-oxide fuel cells. *Science.* **312**, 254–257 (2006).

R34: Zener, C. Interaction between the d-shells in the transition metals. II. Ferromagnetic compounds of manganese with perovskite structure. *Phys. Rev.* **82**, 403 (1951).

R35: Sengodan, S., Ahn, S., Shin, J., & Kim, G. Oxidation–reduction behavior of $\text{La}_{0.8}\text{Sr}_{0.2}\text{Sc}_y\text{Mn}_{1-y}\text{O}_{3\pm\delta}$ ($y = 0.2, 0.3, 0.4$): Defect structure, thermodynamic and electrical properties. *Solid State Ion.* **228**, 25–31 (2012).

Comment 7. Some typos: in the part of Transmission electron microscopy analysis. “0.395 nm (**Figure 3a**) and 0.634 nm (**Figure 3c**)” and “the simple perovskite (**Figure 3b**) and R-P perovskite (**Figure 3d**)” should be changed to “0.395 nm (**Figure 3a**) and 0.634 nm (**Figure 3d**)” and “the simple perovskite (**Figure 3b**) and R-P perovskite (**Figure 3e**)”.

Response to Comment 7: We thank the reviewer for notifying the typos in our manuscript. We made corrections for the transmission electron microscopy (TEM) analysis part and colored as red color below. Moreover, the authors also thoroughly checked whether there are more typos and/or mistakes in our manuscript and found some things that needs to be fixed (the caption for **Supplementary Figure 4** (was **Supplementary Figure 3**)).

Revised text in the revised manuscript:

Results: Transmission electron microscopy analysis

[Original text-(1)]

From the high-resolution TEM images and corresponding fast-Fourier transformed (FFT) patterns, the lattice spaces between planes of A-PBSF30 and R-PBSF30 are 0.395 nm (**Figure 3a**) and 0.634 nm (**Figure 3c**), ...

[Revised text-(1)]

From the high-resolution TEM images and corresponding fast-Fourier transformed (FFT) patterns, the lattice spaces between planes of A-PBSF30 and R-PBSF30 are 0.395 nm (**Fig. 3a**) and 0.634 nm (**Fig. 3d**), ...

[Original text-(2)]

The locations of cations are well-matched with the simple perovskite (**Figure 3b**) and R-P perovskite (**Figure 3d**) because the atomic column intensity is proportional to ...

[Revised text-(2)]

The locations of cations are well-matched with the simple perovskite (**Fig. 3b**) and R-P perovskite (**Fig. 3e**) because the atomic column intensity is proportional to ...

[Original text-(3)]

Before examining the electro-catalytic effect of the *in-situ* exsolved Fe metal particles, an explicit comparison of exsolved particle and surface distribution for R-PBSF00, R-PBSF30, and R-PBSF50 samples were presented in scanning electron microscope (SEM) images (**Figure 3e, Figure 3f, Supplementary Figure 6, and Supplementary Figure 7**).

[Revised text-(3)]

Before examining the electro-catalytic effect of the *in-situ* exsolved Fe metal particles, an explicit comparison of exsolved particle and surface distribution for R-PBSF00, R-PBSF30, and R-PBSF50 samples were presented in scanning electron microscope (SEM) images (**Fig. 3c and 3f, Supplementary Figs. 9 and 10**).

Answers to Reviewer #3

General Comments:

This manuscript deals with studying the oxygen vacancy formation energies in perovskites as a potential descriptor for predicting the amount of exsolved metallic particles upon strong reduction of the material. To achieve as much exsolved metal particles as possible, the authors claim that complete phase reconstruction from simple to Ruddlesden-Popper perovskite is the most suitable pathway. Their approach is a combination of DFT calculations (to predict the most suitable composition for complete phase reconstruction) and in-situ X-ray diffraction (to map the phase diagram, where the observed phase evolution is plotted as a function of A-site doping). Furthermore, the material with the most promising exsolution behaviour is tested for its suitability as an SOFC electrode. The DFT and in-situ XRD part are sound and especially the obtained phase diagram is a strong result. However, what I completely missed is a discussion on why the oxygen vacancy formation enthalpy determines the phase transition behaviour. In this question, the reader is left entirely to his or her own thoughts, which should not be the case with a paper in *Nature communications*. There definitely needs to be more detailed explanation and discussion before the paper can be recommended for publication. Moreover, I unfortunately have to say that the electrochemical part of the paper is rather poor and needs major revisions. I am sorry for this harsh judgement, since I principally like the approach of the authors, but there are yet too much flaws in the manuscript and I therefore recommend that it is reconsidered in a largely revised version.

Response to General Comments: First of all, we sincerely appreciate the reviewer for providing meaningful and thoughtful comments on our work. We are also thankful that the reviewer likes the approach of this work. The detailed comments were greatly helpful to further strengthen the quality of this work. We have revised the manuscript to address the reviewer's opinion as much as possible and the point-by-point response for each comment are listed as below:

Detailed comments and criticism:

Comment 1. The definition of the enthalpy limits between the different phase regions given in Fig.1 is crucial for a correct prediction of exsolution behaviour, therefore it needs more discussion. Where do the values given as limits come from? This appears to be especially important, as three or four of the calculated values are quite close at these limits. Hence, slight shifts in the defined limits or small errors from the calculation may lead to wrong predictions. I suggest to compare the obtained oxygen vacancy formation enthalpies with several literature data. There is already quite some comparable data available – e.g. on (La,Sr)CoO₃ or (La,Sr)MnO₃ based materials. Moreover, quantitatively knowing the errors/uncertainties of the calculation results would be advantageous to being able judging the reliability of the calculated results. I am aware that commonly no errors are given for DFT results, since very often an accuracy of some 0.1 eV is sufficient for supporting or disproving a hypothesis, which was gained by interpretation of experimental results. Here, however, the accuracy of the oxygen vacancy formation energies appears to be very important, since the DFT results are very close at the borderline that decides about which behaviour to expect.

Response to Comment 1: We thank the reviewer for providing us insightful comment. We completely agree with the reviewer's comment that we need to clearly justify the physical meaning of two criteria (phase

decomposition and phase reconstruction) with regards to the oxygen vacancy formation energy. Therefore, as the reviewer pointed out, we validated the physical meaning of both criteria by additional Gibbs free energy for oxygen vacancy formation energy (G_{vf-O}) calculations that contains real experimental conditions. Also, in addressing the reviewer's comment concerning density functional theory (DFT) errors in the borderline cases, we also checked that there was an internal error when using the generalized gradient approximation (GGA) functional approach for DFT calculations^{R4}. Given that the GGA functional over-estimates the O-O bonding of oxygen molecules versus O-T (T = transition metal) bonding in metal oxides, we corrected the related error using the experimentally measured formation energies of metal oxides to calculate more exact G_{vf-O} .

In this manuscript, we initially compared the E_{vf-O} from the AO and BO_2 layers in $Pr_{0.5}(Ba/Sr)_{0.5}TO_{3-\delta}$ (T = Mn, Fe, Co, and Ni) and proposed three possible ranges by two criteria (phase decomposition and phase reconstruction occurring under reducing condition): (i) Phase decomposition occurs under reducing condition when the A-site (AO layer) $E_{vf-O} < 1.5$ eV, (ii) Phase reconstruction to Ruddlesden-Popper (R-P) perovskite occurs under reducing condition when the A-site $E_{vf-O} > 1.5$ eV and 1.6 eV $<$ the B-site (BO_2 layer) $E_{vf-O} < 2.8$ eV, and (iii) Phase reconstruction to layered perovskite and/or the phase remains as simple perovskite when the A-site $E_{vf-O} > 1.5$ eV and the B-site $E_{vf-O} > 2.8$ eV. However, after careful consideration, we thought that the calculated E_{vf-O} values could not provide the absolute standard for the phase decomposition and/or phase reconstruction (**Fig. 1a** in the original manuscript) because these values were relatively identified values that did not contain temperature and oxygen partial pressure factors. To contain the temperature and oxygen partial pressure terms in the E_{vf-O} calculations, we additionally calculated the Gibbs free energy difference oxygen vacancy formation (G_{vf-O}) by using the Gibbs free energies of oxygen, hydrogen, and water molecules. For reference, the abbreviations for eight samples used for the G_{vf-O} calculations are given in **Table R1** to remove confusion concerning sample information. The re-calculated G_{vf-O} for eight samples from the surface AO layer (green bar) and BO_2 layer (purple bar) are displayed in revised **Fig. 1a** (in the response letter). Unlike the E_{vf-O} calculations, since the G_{vf-O} contains temperature and oxygen partial pressure terms, we changed the standard value (borderline) for the phase reconstruction behavior: (i) Phase decomposition occurs under reducing condition when the A-site (surface AO layer) $G_{vf-O} < 0$ eV, (ii) Phase reconstruction to Ruddlesden-Popper (R-P) perovskite occurs under reducing condition when the A-site $G_{vf-O} > 0$ eV and -1.2 eV $<$ the B-site (BO_2 layer) $G_{vf-O} < 0$ eV, and (iii) Phase reconstruction to layered perovskite and/or the phase remains as simple perovskite when the A-site $G_{vf-O} > 0$ eV and the B-site $G_{vf-O} > 0$ eV.

The detailed explanation for the G_{vf-O} calculations from the surface AO and BO_2 layers in $Pr_{0.5}(Ba/Sr)_{0.5}TO_{3-\delta}$ (T = Mn, Fe, Co, and Ni) are as follows:

(1) Phase decomposition at 750 °C and $p(O_2) : 10^{-9}$ atm (Surface AO layer)

The phase for $Pr_{0.5}(Ba/Sr)_{0.5}CoO_{3-\delta}$ -based electrodes are reported to be decomposed at 700 °C under $p(O_2) < 10^{-6}$ atm ($p(O_2)$ similar to argon gas)^{R1}. Considering this experimental data, we re-calculated the G_{vf-O} values from the surface AO layer of $Pr_{0.5}(Ba/Sr)_{0.5}TO_{3-\delta}$ (T = Mn, Fe, Co, and Ni) at 750 °C and $p(O_2) : 10^{-9}$ atm (reducing condition) by using the following equations^{R2} (abbreviations given in **Table R1**):

Equation R1 and Equation R2:

$$G_{vf-O} = E_{perov-defect} - E_{perov} + \frac{1}{2}\mu_{O_2} \quad (R1)$$

$$\mu_{O_2} = E_{O_2(g)}^{DFT} + E_{O_2(g)}^{ZPE} + E_{O_2(g)}^{correction} - TS_{O_2(g)} + k_B T \ln\left(\frac{p_{O_2}}{p_0}\right) \quad (R2)$$

The $E_{perov-defect}$ and E_{perov} are the total energies of PrO-terminated (001) perovskite slab model with and without the oxygen vacancy, respectively. The μ_{O_2} is the Gibbs free energy of the di-atomic oxygen molecule. The $E_{O_2(g)}^{DFT}$ is the gas phase energy of ground state triplet O_2 molecule. The zero point energy ($E_{O_2(g)}^{ZPE}$) and was extracted from the previous calculated value^{R3}. The standard entropy of gas phase oxygen ($S_{O_2(g)}$) was obtained from National Institute of Standards and Technology Chemistry Web-Book (<http://webbook.nist.gov/chemistry>). Moreover, the correction energy of oxygen molecule ($E_{O_2(g)}^{correction}$) was added to reconcile the E_{vf-O} differences between the results achieved *via* computational method (generalized gradient approximation (GGA) functional) and real experimental results^{R4}.

With the re-calculated G_{vf-O} values from the surface AO layer, we can firstly screen the structurally unstable materials at 750 °C under reducing condition. Four materials (PBN, PSN, PBC, and PSC) demonstrated spontaneous oxygen vacancy formation ($G_{vf-O} < 0$) at the surface AO-terminated layers at 750 °C and $p(O_2) : 10^{-9}$ atm (reducing condition), implying phase decomposition at this specified condition. However, the other four materials (PBM, PSM, PBF, and PSF) displayed unspontaneous oxygen vacancy formation ($G_{vf-O} > 0$) at the surface AO-terminated layers at 750 °C and $p(O_2) : 10^{-9}$ atm (reducing condition) (Revised Fig. 1a).

(2) Phase reconstruction at 750 °C, $p(O_2) : 10^{-9}$ atm, $p(H_2) = 0.1$ atm, and $p(H_2O) = 0.01$ atm (BO_2 layer).

The phase reconstruction to Ruddlesden-Popper (R-P) perovskite or layered perovskite occurs by the reduction of BO_2 layers in the reduction atmosphere. Because the formation of oxygen vacancies at the BO_2 layer would include the hydrogen oxidation reaction (HOR, $O_{lattice} + H_2(g) \leftrightarrow H_2O(g)$) at the surface and the oxygen vacancy diffusion toward the BO_2 layer, we also calculated the G_{vf-O} from the BO_2 layer at the above specified condition (reducing condition) by using the following equations:

Equation R3, Equation R4, and Equation R5:

$$G_{vf-O} = E_{perov-defect} - E_{perov} + (\mu_{H_2O} - \mu_{H_2}) + E_a^{O_{vac} diffusion} \quad (R3)$$

$$\mu_{H_2O} = \left(\Delta H_{H_2O}^{exp} + E_{H_2(g)}^{DFT} + E_{H_2(g)}^{ZPE} + \frac{1}{2} (E_{O_2(g)}^{DFT} + E_{O_2(g)}^{ZPE} + E_{O_2(g)}^{correction}) \right) - TS_{H_2O(g)} + k_B T \ln\left(\frac{p_{H_2O}}{p_0}\right) \quad (R4)$$

$$\mu_{H_2} = E_{H_2(g)}^{DFT} + E_{H_2(g)}^{ZPE} - TS_{H_2(g)} + k_B T \ln\left(\frac{p_{H_2}}{p_0}\right) \quad (R5)$$

The zero-point energies were extracted from the previously calculated values^{R3}. The standard entropies of gas phase molecules, as well as the formation enthalpy of water ($(\Delta H_{H_2O}^{exp} : -286.41 \text{ kJ/mol})$), were obtained

from National Institute of Standards and Technology Chemistry Web-Book (<http://webbook.nist.gov/chemistry>). The activation energy of oxygen vacancy diffusion ($E_a^{O_{vac} \text{ diffusion}}$: 0.95 eV) was calculated from the electrochemical measurements (Arrhenius plot of area-specific resistance) of $\text{Pr}_{0.4}\text{Sr}_{0.6}\text{Fe}_{0.875}\text{Mo}_{0.125}\text{O}_{3-\delta}$ (PSFM) under H_2 condition^{R5}. Moreover, it was reported that the PBM perovskite undergoes phase transition to layered perovskite under specified reducing condition (At 750 °C, $p(\text{O}_2) : 10^{-9}$ atm, $p(\text{H}_2) = 0.1$ atm, and $p(\text{H}_2\text{O}) = 0.01$ atm). Thus, the $G_{\text{v-f-O}}$ for eight samples from the BO_2 layer was recalculated at the above specified condition to find the appropriate samples that could demonstrate phase reconstruction to R-P perovskite.

In summary, we additionally calculated the $G_{\text{v-f-O}}$ of eight samples (in **Table R1**) from the surface AO and BO_2 layers to include the main factors for the oxygen vacancy formation (in this case, temperature and oxygen partial pressure). In the presented calculations, the borderlines for two criteria (phase decomposition and phase reconstruction) were both set to the $G_{\text{v-f-O}}$ of 0 eV for checking the spontaneity for two criteria (Revised **Fig. 1a**). Furthermore, by calculation of spontaneous phase decomposition and/or reconstruction temperature where the $G_{\text{v-f-O}}$ value from surface AO layer and BO_2 layer becomes zero (**Fig. R1**), we can approximately determine whether the sample is stable at a specified temperature under reducing condition.

[References for the revision]

R1: Choi, S. *et al.* Highly efficient and robust cathode materials for low-temperature solid oxide fuel cells: $\text{PrBa}_{0.5}\text{Sr}_{0.5}\text{Co}_{2-x}\text{Fe}_x\text{O}_{5+\delta}$. *Sci. Rep.* **3**, 1–6 (2013).

R2: Lee, D. *et al.* Oxygen surface exchange kinetics and stability of $(\text{La,Sr})_2\text{CoO}_{4+\delta}/\text{La}_{1-x}\text{Sr}_x\text{MO}_{3-\delta}$ (M= Co and Fe) hetero-interfaces at intermediate temperatures. *J. Mater. Chem. A.* **3**, 2144–2157 (2015).

R3: Nørskov, J. K. *et al.* Origin of the overpotential for oxygen reduction at a fuel-cell cathode. *J. Phys. Chem. B.* **108**, 17886–17892 (2004).

R4: Wang, L., Maxisch, T., & Ceder, G. Oxidation energies of transition metal oxides within the GGA + U framework. *Phys. Rev. B.* **73**, 195107 (2006).

R5: Zhang, D. *et al.* Preparation and characterization of a redox-stable $\text{Pr}_{0.4}\text{Sr}_{0.6}\text{Fe}_{0.875}\text{Mo}_{0.125}\text{O}_{3-\delta}$ material as a novel symmetrical electrode for solid oxide cell application. *Int. J. Hydrog. Energy.* **45**, 21825–21835 (2020).

R6: Sengodan, S. *et al.* Layered oxygen-deficient double perovskite as an efficient and stable anode for direct hydrocarbon solid oxide fuel cells. *Nat. Mater.* **14**, 205–209 (2015).

[Table R1]

Table R1. Chemical composition and abbreviation of specimens used for the density functional theory (DFT) calculations.

Composition	Abbreviation
$\text{Pr}_{0.5}\text{Ba}_{0.5}\text{MnO}_{3-\delta}$	PBM
$\text{Pr}_{0.5}\text{Sr}_{0.5}\text{MnO}_{3-\delta}$	PSM
$\text{Pr}_{0.5}\text{Ba}_{0.5}\text{FeO}_{3-\delta}$	PBF (A-PBSF00 in the manuscript)

$\text{Pr}_{0.5}\text{Sr}_{0.5}\text{FeO}_{3-\delta}$	PSF (A-PBSF50 in the manuscript)
$\text{Pr}_{0.5}\text{Ba}_{0.5}\text{CoO}_{3-\delta}$	PBC
$\text{Pr}_{0.5}\text{Sr}_{0.5}\text{CoO}_{3-\delta}$	PSC
$\text{Pr}_{0.5}\text{Ba}_{0.5}\text{NiO}_{3-\delta}$	PBN
$\text{Pr}_{0.5}\text{Sr}_{0.5}\text{NiO}_{3-\delta}$	PSN

[Figure R1]

Figure R1. (a) Spontaneous phase decomposition temperature where the Gibbs free energy for oxygen vacancy formation (G_{vf-O}) at the surface AO layer becomes zero under $p(\text{O}_2)$: 10^{-9} atm. (b) Spontaneous reduction temperature where the G_{vf-O} at the BO_2 layer becomes zero under $p(\text{O}_2)$: 10^{-9} atm, P_{H_2} : 0.1 atm, and $P_{\text{H}_2\text{O}}$: 0.01 atm.

Added text in the revised manuscript:

Methods: Computational methods

Furthermore, the Gibbs free energy for oxygen vacancy formation of eight samples were calculated to include the temperature and oxygen partial pressure factors in the E_{vf-O} calculations. The equations used for the G_{vf-O} calculations from the surface AO and BO_2 layers in $\text{Pr}_{0.5}(\text{Ba}/\text{Sr})_{0.5}\text{TO}_{3-\delta}$ ($T = \text{Mn, Fe, Co, and Ni}$) are as follows:

$$G_{vf-O}(\text{AO layer}) = E_{\text{perov-defect}} - E_{\text{perov}} + \frac{1}{2}\mu_{\text{O}_2}$$

$$\mu_{\text{O}_2} = E_{\text{O}_2(g)}^{\text{DFT}} + E_{\text{O}_2(g)}^{\text{ZPE}} + E_{\text{O}_2(g)}^{\text{correction}} - TS_{\text{O}_2(g)} + k_B T \ln\left(\frac{P_{\text{O}_2}}{P_0}\right)$$

$$G_{vf-O}(\text{BO}_2 \text{ layer}) = E_{\text{perov-defect}} - E_{\text{perov}} + (\mu_{\text{H}_2\text{O}} - \mu_{\text{H}_2}) + E_a^{\text{Ovac diffusion}}$$

$$\mu_{\text{H}_2\text{O}} = \left(\Delta H_{\text{H}_2\text{O}}^{\text{exp}} + E_{\text{H}_2(g)}^{\text{DFT}} + E_{\text{H}_2(g)}^{\text{ZPE}} + \frac{1}{2}(E_{\text{O}_2(g)}^{\text{DFT}} + E_{\text{O}_2(g)}^{\text{ZPE}} + E_{\text{O}_2(g)}^{\text{correction}})\right) - TS_{\text{H}_2\text{O}(g)} + k_B T \ln\left(\frac{P_{\text{H}_2\text{O}}}{P_0}\right)$$

$$\mu_{\text{H}_2} = E_{\text{H}_2(g)}^{\text{DFT}} + E_{\text{H}_2(g)}^{\text{ZPE}} - TS_{\text{H}_2(g)} + k_B T \ln\left(\frac{P_{\text{H}_2}}{P_0}\right)$$

The $E_{\text{perov-defect}}$ and E_{perov} are the total energies of PrO -terminated (001) perovskite slab model with and without the oxygen vacancy, respectively. The μ_{O_2} , μ_{H_2} , and $\mu_{\text{H}_2\text{O}}$ are the Gibbs free energy of diatomic oxygen molecule, hydrogen, and water molecule, respectively. The $E_{\text{O}_2(g)}^{\text{DFT}}$ and $E_{\text{H}_2(g)}^{\text{DFT}}$ is the gas

phase energy of ground state triplet O₂ molecule and hydrogen molecule, respectively. The zero point energies of oxygen molecule ($E_{O_2(g)}^{ZPE}$) and hydrogen molecule ($E_{H_2(g)}^{ZPE}$) were extracted from the previous calculated value⁵⁹. The standard entropy of gas phase oxygen ($S_{O_2(g)}$) was obtained from National Institute of Standards and Technology Chemistry Web-Book (<http://webbook.nist.gov/chemistry>). Moreover, the correction energy of oxygen molecule ($E_{O_2(g)}^{correction}$) was added to reconcile the $E_{v-f,O}$ differences between the results achieved *via* computational method (generalized gradient approximation (GGA) functional) and real experimental results⁶⁰. The p(O₂), p(H₂), and p(H₂O) values are 10⁻⁹, 0.1, and 0.01 atm, respectively. Under this specified condition (reducing condition), we assumed that the reduction of BO₂ layer occurred *via* two elementary steps: surface hydrogen oxidation reaction ($O_{lattice} + H_2(g) \leftrightarrow H_2O(g)$) and oxygen vacancy diffusion toward the BO₂ layer. The activation energy of oxygen vacancy diffusion ($E_a^{O_{vac} diffusion}$: 0.95 eV) was calculated from the electrochemical measurements (Arrhenius plot of area specific resistance) of Pr_{0.4}Sr_{0.6}Fe_{0.875}Mo_{0.125}O_{3-δ} (PSFM) material under H₂ condition⁶¹.

Added references in the revised manuscript:

59: Nørskov, J. K. *et al.* Origin of the overpotential for oxygen reduction at a fuel-cell cathode. *J. Phys. Chem. B.* **108**, 17886–17892 (2004).

60: Wang, L., Maxisch, T., & Ceder, G. Oxidation energies of transition metal oxides within the GGA + U framework. *Phys. Rev. B.* **73**, 195107 (2006).

61: Zhang, D. *et al.* Preparation and characterization of a redox-stable Pr_{0.4}Sr_{0.6}Fe_{0.875}Mo_{0.125}O_{3-δ} material as a novel symmetrical electrode for solid oxide cell application. *Int. J. Hydrog. Energy.* **45**, 21825–21835 (2020).

Added figure in the revised manuscript:

[Supplementary Figure 1]

Supplementary Figure 1. Calculated oxygen vacancy formation energies of Pr_{0.5}(Ba/Sr)_{0.5}T_{0.3-δ} (T = Mn, Fe, Co, and Ni) from AO (green bar) and BO₂ (purple bar) networks and the predicted phase change under reducing condition. Note that the Gibbs free energy for the oxygen vacancy formation ($G_{v-f,O}$) was additionally calculated to contain the temperature and oxygen partial pressure factors in the $E_{v-f,O}$ calculations.

Revised figure in the revised manuscript:

[Original figure]

Figure 1. (a) Calculated oxygen vacancy formation energies of $\text{Pr}_{0.5}(\text{Ba}/\text{Sr})_{0.5}\text{TO}_{3-\delta}$ ($T = \text{Mn, Fe, Co, and Ni}$) from AO (green bar) and BO_2 (purple bar) networks and the predicted phase change under reducing condition. (b) Schematic of the most stable structure configurations of $\text{Pr}_{0.5}(\text{Ba}/\text{Sr})_{0.5}\text{TO}_{3-\delta}$ ($T = \text{Mn, Fe, Co, and Ni}$) slab models used for the calculations of oxygen vacancy formation energy values from AO and BO_2 networks.

[Revised figure]

Figure 1. (a) Calculated Gibbs free energy for oxygen vacancy formation (G_{vf-O}) of $\text{Pr}_{0.5}(\text{Ba/Sr})_{0.5}\text{TO}_{3-\delta}$ (T = Mn, Fe, Co, and Ni) from the surface AO (green bar) and BO_2 (purple bar) networks and the predicted phase change under reducing condition. (b) Schematic of the most stable structure configurations of $\text{Pr}_{0.5}(\text{Ba/Sr})_{0.5}\text{TO}_{3-\delta}$ (T = Mn, Fe, Co, and Ni) slab models used for the calculations of G_{vf-O} values from the surface AO and BO_2 networks.

Revised text in the revised manuscript:

Abstract

[Original text]

Herein, the oxygen vacancy formation energies (E_{vf-O}) from PrO and TO_2 in $\text{Pr}_{0.5}(\text{Ba/Sr})_{0.5}\text{TO}_{3-\delta}$ (T = Mn, Fe, Co, and Ni) are proposed as the important factor in ...

[Revised text]

Herein, the Gibbs free energy for oxygen vacancy formation (G_{vf-O}) from PrO and TO_2 in $\text{Pr}_{0.5}(\text{Ba/Sr})_{0.5}\text{TO}_{3-\delta}$ (T = Mn, Fe, Co, and Ni) are proposed as the important factor in ...

Introduction

[Original text]

Here, we systematically calculated the oxygen vacancy formation energies (E_{vf-O}) of perovskite oxides with various cations to investigate the unprecedented factor affecting the phase reconstruction. The type of phase reconstruction could be predicted with the E_{vf-O} value from PrO and TO_2 networks in $\text{Pr}_{0.5}(\text{Ba/Sr})_{0.5}\text{TO}_{3-\delta}$ (T =

Mn, Fe, Co, and Ni), in which the most appropriate cations for the complete reconstruction to R-P perovskite were determined.

[Revised text]

Here, we systematically calculated the **Gibbs free energy for oxygen vacancy formation (G_{vf-O})** of perovskite oxides with various cations to investigate the unprecedented factor affecting the phase reconstruction. The type of phase reconstruction could be predicted with the G_{vf-O} value from PrO and TO_2 networks in $Pr_{0.5}(Ba/Sr)_{0.5}TO_{3-\delta}$ (T = Mn, Fe, Co, and Ni), in which the most appropriate cations for the complete reconstruction to R-P perovskite were determined.

Results: Density functional theory calculations

[Original text-(1)]

To determine the unexplored factor for the phase reconstruction for the first time, the oxygen vacancy formation energies (E_{vf-O}) from AO (A-site) and BO_2 (B-site) networks were calculated for $Pr_{0.5}Ba_{0.5}TO_{3-\delta}$ and $Pr_{0.5}Sr_{0.5}TO_{3-\delta}$ (T = Mn, Fe, Co, and Ni) perovskite oxides (**Figure 1**)³¹⁻³⁴.

[Revised text-(1)]

To determine the unexplored factor for the phase reconstruction for the first time, **the Gibbs free energy for oxygen vacancy formation (G_{vf-O})** and the oxygen vacancy formation energies (E_{vf-O}) from **the surface** AO (A-site) and BO_2 (B-site) networks were calculated for $Pr_{0.5}Ba_{0.5}TO_{3-\delta}$ and $Pr_{0.5}Sr_{0.5}TO_{3-\delta}$ (T = Mn, Fe, Co, and Ni) perovskite oxides (**Fig. 1 and Supplementary Fig. 1**)³¹⁻³⁴.

[Original text-(2)]

For the perovskite oxides to undergo phase reconstruction without phase decomposition under reducing condition, the A-site E_{vf-O} value should be higher than 1.5 eV. Moreover, the B-site E_{vf-O} value would be an important factor for determining the type of phase reconstruction. For instance, the B-site E_{vf-O} should be in the range of about 1.6 to 2.8 eV to demonstrate phase reconstruction to R-P perovskite in the reduction environment.

[Revised text-(2)]

For the perovskite oxides to undergo phase reconstruction without phase decomposition under reducing condition, the A-site G_{vf-O} value should be **positive (A-site $G_{vf-O} > 0$)**. Moreover, the B-site G_{vf-O} value would be an important factor for determining the type of phase reconstruction. For instance, the B-site G_{vf-O} should be in the range of about **- 1.2 to 0 eV (- 1.2 < B-site $G_{vf-O} < 0$)** to demonstrate phase reconstruction to R-P perovskite in the reduction environment.

Conclusion

[Original text]

In summary, this study successfully calculated E_{VFO} value from PrO and TO_2 in $\text{Pr}_{0.5}(\text{Ba/Sr})_{0.5}\text{TO}_{3-\delta}$ ($T = \text{Mn, Fe, Co, and Ni}$) as the key factor for identifying the type of the phase reconstruction.

[Revised text]

In summary, this study successfully calculated G_{VFO} value from PrO and TO_2 in $\text{Pr}_{0.5}(\text{Ba/Sr})_{0.5}\text{TO}_{3-\delta}$ ($T = \text{Mn, Fe, Co, and Ni}$) as the key factor for identifying the type of the phase reconstruction.

Comment 2. Owing to my opinion several of the Figures from the supporting info should be moved to the main text. As far as I remember correctly, Nature Communications allows up to 7 display items. So there should be plenty of space available.

Response to Comment 2: We thank the reviewer for the helpful comment. We checked the “manuscript checklist” for *Nature Communications* and found that *Nature communication* allows up to 10 display items (no more than 10 total). First, we moved the X-ray photoelectron spectra (XPS) data in **Supplementary Fig. 10** to the main figure to support the X-ray absorption fine structure (XAFS) measurement data that we newly attached in **Fig. 4**. Furthermore, we added a schematic figure to help readers easily understand about this work (**Fig. 6**). From the addition of these two figures, the number of main figures is 6 total. This comment gave us a great chance to strengthen the quality of the figures explaining the main parts of this work.

Added figures in the revised manuscript:

Figure 4. (a – e) Oxidation state characterization. (a – b) Fe K-edge X-ray absorption near-edge structure

(XANES) spectra of $\text{Pr}_{0.5}\text{Ba}_{0.2}\text{Sr}_{0.3}\text{FeO}_{3-\delta}$ (A-PBSF30), $(\text{Pr}_{0.5}\text{Ba}_{0.2}\text{Sr}_{0.3})_2\text{FeO}_{4+\delta}$ (R-PBSF30) with two references (Fe foil and Fe_2O_3). (c) Fourier-transformed Fe K-edge extended X-ray absorption fine structure (EXAFS) spectra of A-PBSF30 and R-PBSF30. (d – e) X-ray photoelectron spectra (XPS) of Fe $2p_{1/2}$ for (d) A-PBSF30 and (e) R-PBSF30.

Figure 6. Schematic illustration of the fuel electrode side of $\text{Pr}_{0.5}\text{Ba}_{0.5-x}\text{Sr}_x\text{FeO}_{3-\delta}$ ($x = 0, 0.3, \text{ and } 0.5$) symmetrical cells and its relation to electrochemical performances.

Revised text in the revised manuscript:

Results: Electrochemical performance evaluation

[Original text]

It is noteworthy that *in-situ* exsolution of well-dispersed Fe metal particles after complete phase reconstruction to R-P perovskite matrix acts as catalysts with promising electro-catalytic activity, leading to outstanding electrochemical performances in various applications.

[Revised text]

It is noteworthy that *in-situ* exsolution of well-dispersed Fe metal particles after complete phase reconstruction to R-P perovskite matrix acts as catalysts with promising electro-catalytic activity (**Fig. 6**), leading to outstanding electrochemical performances in various applications.

Comment 3. I do not agree with the fitting approach of the Fe XPS spectra. First of all, measuring only the Fe $2p_{1/2}$ line provides not sufficient information for deciding the oxidation state of iron, since also the shake-up features between Fe $2p_{1/2}$ and Fe $2p_{3/2}$ peak need to be considered to draw such conclusions.[1] Second, identification of Fe^{4+} only from XPS is daring – in perovskites for example, localization of electron holes at the Fe-O molecular orbital is more likely as already confirmed by XAS experiments. [2]

Response to Comment 3: We thank the reviewer for the valuable comment that could develop this manuscript. We also agree about the reviewer's thoughtful comment that only providing the X-ray photoelectron spectroscopy (XPS) plot for only Fe $2p_{1/2}$ is not sufficient for deciding the oxidation state of iron^{R36,R37}. Since the X-ray photoelectron spectroscopy (XPS) is only limited to the surface analysis of much small area and identification of Fe^{4+} only from XPS is daring, we additionally performed X-ray absorption fine structure (XAFS) measurements to accurately confirm the oxidation state of iron in bulk scale (**Fig. 4**). Consequently, as indicated from the magnified Fe K-edge X-ray absorption near edge structure (XANES) spectra at $f''(E) = \text{maximum}$ (**Fig. 4b**), Fe^{4+} could be identified from the $\text{Pr}_{0.5}\text{Ba}_{0.2}\text{Sr}_{0.3}\text{FeO}_{3-\delta}$ (A-PBSF30)

material because the Fe K-edge energy position for A-PBSF30 is higher than that of Fe_2O_3 reference (Fe^{3+}). In addition, one of the main points of this manuscript is to confirm the Fe metal exsolution after the phase reconstruction from A-PBSF30 to $(\text{Pr}_{0.5}\text{Ba}_{0.2}\text{Sr}_{0.3})_2\text{FeO}_{4+\delta}$ - Fe metal (R-PBSF30) under reducing condition. As demonstrated in **Fig. 4c**, the intensity for Fe-O shell decreases while the intensity for Fe-Fe shell increases much after the phase reconstruction from A-PBSF30 to R-PBSF30, verifying much exsolution of Fe metal from the perovskite parent in the reduction atmosphere.

[References for the revision]

R36: Descostes, M., Mercier, F., Thromat, N., Beaucaire, C., & Gautier-Soyer, M. Use of XPS in the determination of chemical environment and oxidation state of iron and sulfur samples: constitution of a data basis in binding energies for Fe and S reference compounds and applications to the evidence of surface species of an oxidized pyrite in a carbonate medium. *Appl. Surf. Sci.* **165**, 288–302 (2000).

R37: Mueller, D. N., Machala, M. L., Bluhm, H., & Chueh, W. C. Redox activity of surface oxygen anions in oxygen-deficient perovskite oxides during electrochemical reactions. *Nat. Commun.* **6**, 1–8 (2015).

Added figure in the revised manuscript:

Figure 4. (a – e) Oxidation state characterization. (a – b) Fe K-edge X-ray absorption near-edge structure (XANES) spectra of $\text{Pr}_{0.5}\text{Ba}_{0.2}\text{Sr}_{0.3}\text{FeO}_{3-\delta}$ (A-PBSF30), $(\text{Pr}_{0.5}\text{Ba}_{0.2}\text{Sr}_{0.3})_2\text{FeO}_{4+\delta}$ (R-PBSF30) with two references (Fe foil and Fe_2O_3). (c) Fourier-transformed Fe K-edge extended X-ray absorption fine structure

(EXAFS) spectra of A-PBSF30 and R-PBSF30. (d – e) X-ray photoelectron spectra (XPS) of Fe 2p_{1/2} for (d) A-PBSF30 and (e) R-PBSF30.

Revised text in the revised manuscript:

Results: Examination and characterization of exsolved particle size

[Original text-(1)]

Furthermore, The presence of Fe⁰ 2p_{1/2} peak for only R-PBSF30 from X-ray photoelectron spectroscopy (XPS) measurements confirm the exsolution of Fe metal onto the surface under reducing condition³⁹, in coincidence with the above experimental results (**Supplementary Figure 10**).

[Revised text-(1)]

Furthermore, noticeable energy shift to the higher energy in X-ray absorption near-edge structure (XANES), much increase in Fe-Fe shell intensity from the Fourier-transformed extended X-ray absorption fine structure (EXAFS) spectra after reduction, and the presence of Fe⁰ 2p_{1/2} peak for only R-PBSF30 from X-ray photoelectron spectroscopy (XPS) measurements confirm the exsolution of Fe metal onto the surface under reducing condition³⁹, in coincidence with the above experimental results (**Fig. 4**).

Added text in the revised manuscript:

Methods: Structural characterization

The X-ray absorption fine structure (XAFS) spectra of Fe K-edge for A-PBSF30, R-PBSF30, and two references (Fe foil and Fe₂O₃ powder) were measured on ionization detectors under fluorescence mode at the Pohang Accelerator Laboratory (PAL, 6D extended XAFS (EXAFS)). The XAFS and Fourier-transformed (FT) EXAFS spectra analysis were performed using the Athena (Demeter) program.

Comment 4. The electrical conductivity of an electrode material alone is not a suitable descriptor for predicting a good electrochemical behaviour. For a sufficiently thick 3D porous electrode with an electronic conductivity higher than the ionic conductivity – which appears to be the case for this material when looking at the relatively high values of total conductivity in Supp. Fig. 11 – the ionic conductivity and the surface reaction resistance are equally important, since the decay length of the electrochemical activity is $\sqrt{(R_{\text{react}}/R_{\text{ion}})}$. [3]

Response to Comment 4: We thank the reviewer for the valuable comment. We definitely agree that the electrical conductivity value, which is the addition of electronic conductivity value and ionic conductivity value, is not alone just a suitable descriptor for predicting the good electrochemical behavior. In addition, even though most perovskite oxide materials with crystalline structure possess much higher electronic conductivity values compared to ionic conductivity values (more than 100 times)^{R38,R39}, the ionic conductivity value should be also considered for 3D porous structure. In this regard, we additionally measured the polarization resistance (or area-specific resistance) of symmetric half-cells and symmetrical full-cells having sufficiently 3D porous electrodes because the polarization resistance value obtained from symmetric half-cell measurement could be a factor to comprehensively explain about the electronic conductivity, ionic

conductivity, and the surface reaction resistance of electrode materials. Hence, we measured the electrochemical impedance spectroscopy (EIS) and plotted the Nyquist plot for (i) $\text{Pr}_{0.5}\text{Ba}_{0.2}\text{Sr}_{0.3}\text{FeO}_{3-\delta}$ (A-PBSF30) and $\text{Pr}_{0.5}\text{Sr}_{0.5}\text{FeO}_{3-\delta}$ (A-PBSF50) symmetric half-cells supplying air to both sides at 800 °C and (ii) A-PBSF30 and A-PBSF50 symmetrical full-cells (Fig. R5). From the area-specific resistance (ASR) difference (or polarization resistance) between symmetrical full-cells (Fig. R5b) and symmetric half-cells (Fig. R5a), it could be elucidated that the maximum power density (P_{max}) difference (electrochemical behavior difference) between A-PBSF30 symmetrical full-cell and A-PBSF50 symmetrical full cell is mostly derived from the electro-catalytic activity of the fuel electrode part.

[References for the revision]

R38: Yoo, S. *et al.* Development of double perovskite compounds as cathode materials for low temperature solid oxide fuel cells. *Angew. Chem. Int. Ed.* **126**, 13280–13283 (2014).

R39: Adler, S. B., Lane, J. A., & Steele, B. C. H. Electrode kinetics of porous mixed-conducting oxygen electrodes. *J. Electrochem. Soc.* **143**, 3554 (1996).

[Figure R5]

Figure R5. (a – b) Electrochemical impedance spectroscopy (EIS) analysis. Impedance spectra of (a) $\text{Pr}_{0.5}\text{Ba}_{0.2}\text{Sr}_{0.3}\text{FeO}_{3-\delta}$ (A-PBSF30) and $\text{Pr}_{0.5}\text{Sr}_{0.5}\text{FeO}_{3-\delta}$ (A-PBSF50) symmetric half-cells supplying air to both sides under open circuit condition and (b) A-PBSF30 and A-PBSF50 symmetrical cells supplying air to the air electrode and H_2 to the fuel electrode under open circuit condition.

Comment 5. The chosen method is largely unsuitable for testing the electrochemical performance of the

electrode material. The reason for this is that the performance of the electrolyte-supported SOFCs manufactured for this purpose is practically only limited by the ohmic resistance of the electrolyte in the investigated temperature range of 700-800°C. This can also be seen very clearly in the practically linear I-V characteristics (Figs. 4 and S14) at low to moderate current densities (the non-linearity at high current density points towards a concentration limitation). Especially at very low currents, the I-V curves hardly show any discernible non-linearity, which could indicate an electrode polarization limitation. This means that the high performance is mainly achieved either by a very good ion-conducting or comparatively thin LSGM electrolyte and therefore no quantitative statement about the electrode performance is permissible. Instead of cell measurements, a characterization of the polarization resistance of the electrodes would be much better suited to compare the investigated material with similar materials in the literature. However, it also needs to be noted that the polarization resistance of porous electrodes strongly depends on their 3D structure (tortuosity, inner surface, etc.). This effect was also completely neglected in the comparison in Fig. 4.

Response to Comment 5: We thank the reviewer for providing constructive comment. The reviewer stated that the electrolyte-supported solid oxide fuel cells (SOFCs) manufactured for the electrochemical performance measurement is practically only limited by the ohmic resistance of the electrolyte in the temperature regime of 700 °C to 800 °C, and the high performance for $\text{Pr}_{0.5}\text{Ba}_{0.2}\text{Sr}_{0.3}\text{FeO}_{3-\delta}$ (A-PBSF30, $x = 0.3$) symmetrical cell ($\text{La}_{0.9}\text{Sr}_{0.1}\text{Ga}_{0.8}\text{Mg}_{0.2}\text{O}_{3-\delta}$ (LSGM) electrolyte-supported symmetrical solid oxide fuel cell) is attributed to very good ion-conducting or comparatively thin LSGM electrolyte. To confirm whether the maximum power density (P_{\max}) difference of similar materials is only affected by the electro-catalytic activity for the SOFC electrodes (electrochemical O_2 reduction and electrochemical H_2 oxidation), we included the electrochemical impedance spectroscopy (EIS) and the corresponding Nyquist plot of symmetrical full-cells that we did not attach in the initial submission (**Supplementary Fig. 17**). Moreover, to verify whether the P_{\max} difference is mostly originated from the SOFC fuel electrode (anode) part, we compared the polarization resistance between A-PBSF30 and $\text{Pr}_{0.5}\text{Sr}_{0.5}\text{FeO}_{3-\delta}$ (A-PBSF50, $x = 0.5$) symmetric half-cells (supplying air to both sides) and symmetrical full-cells. Since the polarization resistance difference between the symmetrical full-cells was much higher than those of symmetric half-cells supplying air to both sides at 800 °C, the P_{\max} difference between A-PBSF30 symmetrical full-cell and A-PBSF50 symmetrical full-cell is mostly derived from the SOFC fuel electrode part (electrochemical H_2 oxidation). The reviewer also mentioned that the polarization resistance of porous electrodes strongly depends on their 3D structure. In this work, since the 3D structure (surface morphology) of all PBSF series sintered at cell fabrication condition (950 °C for 4 hours in air atmosphere) are similar (**Supplementary Fig. 15**), the electrochemical performance and the polarization resistance would be influenced by the electro-catalytic activity of electrodes, not by the surface morphology of 3D-structured porous electrodes.

Added figure in the revised manuscript:

[Supplementary Figure 17]

Supplementary Figure 17. (a – f) Electrochemical impedance spectra (EIS) of the symmetrical cells with $\text{Pr}_{0.5}\text{Ba}_{0.5-x}\text{Sr}_x\text{FeO}_{3-\delta}$ electrodes using humidified H_2 (3% H_2O) as fuel and air as oxidant from 700 °C to 800 °C with intervals of 50 °C: (a) A-PBSF00 ($x = 0$), (b) A-PBSF10 ($x = 0.1$), (c) A-PBSF20 ($x = 0.2$), (d) A-PBSF30 ($x = 0.3$), (e) A-PBSF40 ($x = 0.4$), and (f) A-PBSF50 ($x = 0.5$).

[Figure R5]

Figure R5. (a – b) Electrochemical impedance spectroscopy (EIS) analysis. Impedance spectra of (a) $\text{Pr}_{0.5}\text{Ba}_{0.2}\text{Sr}_{0.3}\text{FeO}_{3-\delta}$ (A-PBSF30) and $\text{Pr}_{0.5}\text{Sr}_{0.5}\text{FeO}_{3-\delta}$ (A-PBSF50) symmetric half-cells supplying air to both sides under open circuit condition and (b) A-PBSF30 and A-PBSF50 symmetrical cells supplying air to the air electrode and H_2 to the fuel electrode under open circuit condition.

Comment 6. Why did the authors decide not to use a Hubbard-U to properly consider effects of electron localisation? Especially for Fe-based perovskites this effect can be important (and at least in the XPS fits the authors do consider localised electrons).

Response to Comment 6: We thank the author for the helpful comment. The generalized gradient approximation (“**GGA**”) based Perdew-Burke-Ernzerhof (PBE) functional along with “**Hubbard U**” (GGA+*U*) approach was utilized in this work. We feel sorry to the reviewer for giving confusion on this part. We stated in the **Methods: Computational methods** part as “GGA+*U* approach was used to correct the self-interaction errors with $U_{\text{eff}} = 4.0$ eV for Fe 3d orbital, $U_{\text{eff}} = 3.3$ eV for Co 3d orbital, $U_{\text{eff}} = 4.0$ eV for Mn 3d orbital, $U_{\text{eff}} = 7.0$ eV for Ni 3d orbital, and $U_{\text{eff}} = 6.0$ eV for Pr 4f orbital^{18,56,57}”. As a result, we used GGA + Hubbard *U* (GGA+*U*) to properly consider the effects of electron localization.

There are a few changes in the affiliation and author contributions, as listed below:

[Affiliation]

Hyunmin Kim^{a,†}, Chaesung Lim^{b,†}, Ohhun Kwon^{c,†}, *Jinkyung Oh^a, Matthew T. Curman^b*, Hu Young Jeong^d, Sihyuk Choi^{e,*}, Jeong Woo Han^{b,*} and Guntae Kim^{a,*}

^a School of Energy and Chemical Engineering, Ulsan National Institute of Science and Technology (UNIST), Ulsan, 44919, Republic of Korea

^b Department of Chemical Engineering, Pohang University of Science and Technology (POSTECH), Pohang, Gyeongbuk 37673, Republic of Korea

^c Department of Chemical and Biomolecular Engineering, University of Pennsylvania, Philadelphia, Pennsylvania 19104, United States of America.

^d Department of Materials Science and Engineering and UNIST Central Research Facilities (UCRF), Ulsan National Institute of Science and Technology (UNIST), Ulsan, 44919, Republic of Korea

^e Department of Mechanical Engineering (Aeronautics, Mechanical and Electronic Convergence Engineering), Kumoh National Institute of Technology, Gyeongbuk 39177, Republic of Korea.

[†] These authors contributed equally to this work.

[Author Contributions]

H.K. and O.K. carried out most of the experimental works and contributed to manuscript writing. C.L. performed DFT calculations. *M.T.C. gave help on additional DFT calculations. J.O. performed the gas chromatography (GC) analysis.* H.Y.J. conducted TEM measurements and analyzed the TEM images. S.C., J.W.H. and G.K. designed the experiments and analyzed the data. All authors contributed to the discussions and analysis of the results regarding the manuscript.

REVIEWERS' COMMENTS

Reviewer #1 (Remarks to the Author):

Most of my comments have been finely replied. I think the manuscript could be accepted.

Reviewer #2 (Remarks to the Author):

The authors have addressed all the questions raised by the reviewer properly. The manuscript has been revised accordingly. Therefore, I suggest accepting this work in the present version.

Reviewer #3 (Remarks to the Author):

The authors have answered all of my questions and comments in an exceptionally thorough manner and have revised their manuscript accordingly. With the manuscript in its present form, all my concerns have been addressed and I believe it is ready for publication.